# Laser-based conversion electron Mössbauer spectroscopy of $^{229}$ThO$_2$

Ricky Elwell[1,10], James E. S. Terhune[1,10], Christian Schneider[1,10], Harry W. T. Morgan[2], Hoang Bao Tran Tan[3,4], Udeshika C. Perera[3], Daniel A. Rehn[4], Marisa C. Alfonso[5], Lars von der Wense[6], Benedict Seiferle[7], Kevin Scharl[7], Peter G. Thirolf[7], Andrei Derevianko[3] & Eric R. Hudson[1,8,9]✉

The exceptionally low-energy $^{229}$Th nuclear isomeric state is expected to provide several new and powerful applications[1,2], including the construction of a robust and portable solid-state nuclear clock[3], perhaps contributing to a redefinition of the second[4], exploration of nuclear superradiance[5,6] and tests of fundamental physics[7–10]. Further, analogous to the capabilities of traditional Mössbauer spectroscopy, the sensitivity of the nucleus to its environment can be used to realize laser Mössbauer spectroscopy and, with it, new types of strain and temperature sensors[3,11] and a new probe of the solid-state environment[12,13], all with excellent sensitivity. However, current models for examining the nuclear transition in a solid require the use of a high-bandgap, vacuum ultraviolet (VUV) transmissive host, severely limiting the applicability of these techniques. Here we report the first, to the authors' knowledge, demonstration of laser-induced conversion electron Mössbauer spectroscopy (CEMS) of the $^{229}$Th isomer in a thin ThO$_2$ sample whose bandgap (approximately 6 eV) is considerably smaller than the nuclear isomeric state energy (8.4 eV). Unlike fluorescence spectroscopy of the $^{229}$Th isomeric transition, this technique is compatible with materials whose bandgap is less than the nuclear transition energy, opening a wider class of systems to study and the potential of a conversion-electron-based nuclear clock.

The recent observation[14,15] of direct laser excitation of the $^{229}$Th isomeric state in high-bandgap crystals has, after almost 50 years of work, opened the door to a laser-accessible nuclear transition and driven rapid progress in the development of solid-state optical clocks. Already, the nuclear transition frequency has been compared with an atomic clock and its linewidth studied[16], and work has begun to optimize the clock performance by developing both a theoretical understanding of nuclear quenching mechanisms[12] and using them to shorten the clock interrogation cycle[17,18]. Similarly, excitation of the nuclear transition in a VUV transmissive thin film has opened the door to integrated-photonic-based nuclear clocks and sensors[19]. Further, new high-bandgap materials have been analysed that could provide considerable simplification of the clock by, for example, doping $^{229}$Th into a nonlinear optical crystal[20] and, by using principles of molecular design, provide a system with the potential for performance orders of magnitude beyond any current or planned optical clock[4,21].

Because the $^{229}$Th nucleus provides a highly controllable system that can be deployed into various hosts, it is expected, in analogy to Mössbauer spectroscopy, that these same techniques can also be used as new probes of the solid-state chemical and nuclear environment. However, in all studies so far, $^{229}$Th nuclear excitation is detected by observing the resulting nuclear fluorescence. This leads to the requirement that the host material has a bandgap larger than the isomeric transition energy ($>E_{iso}$), as the large internal conversion (IC) coefficient of the isomeric state (roughly $10^8$–$10^9$ (refs. 22,23)) extinguishes the nuclear fluorescence. This requirement severely limits the material hosts available for study and, therefore, it is highly desirable to extend the spectroscopy of the isomeric transition to low-bandgap ($<E_{iso}$) environments by means of a new method for nuclear excitation detection.

CEMS, which has been used for decades as an important materials probe[24–26] and was recently used to detect the 12.4-keV nuclear clock transition in $^{45}$Sc (ref. 27), has been proposed for detection of $^{229}$Th nuclear excitation[28,29]. CEMS of the $^{229}$Th isomer uses the fact that, if the nuclear energy is larger than the material bandgap, the IC relaxation process is possible by promoting an electron across the bandgap[23,30]. If this promoted IC electron originates in a shallow enough state in the valence band, the electron can overcome the ionization energy barrier (that is, the difference between the top of the valence band and the vacuum[31]) and emerge from the material surface. By detecting these conversion electrons, nuclear spectroscopy can be recorded and, as a result, the technique of tabletop laser-based nuclear spectroscopy can be extended to low-bandgap materials[28,29] through this laser-based CEMS.

[1]Department of Physics and Astronomy, University of California, Los Angeles, Los Angeles, CA, USA. [2]Department of Chemistry, University of Manchester, Manchester, UK. [3]Department of Physics, University of Nevada, Reno, Reno, NV, USA. [4]Computational Physics Division, Los Alamos National Laboratory, Los Alamos, NM, USA. [5]Eckert & Ziegler Analytics, Inc., Atlanta, GA, USA. [6]Department of Physics, Johannes Gutenberg-Universität Mainz, Mainz, Germany. [7]Faculty of Physics, Ludwig-Maximilians-Universität München, Garching, Germany. [8]Challenge Institute for Quantum Computation, University of California, Los Angeles, Los Angeles, CA, USA. [9]Center for Quantum Science and Engineering, University of California, Los Angeles, Los Angeles, CA, USA. [10]These authors contributed equally: Ricky Elwell, James E. S. Terhune, Christian Schneider. ✉e-mail: eric.hudson@ucla.edu

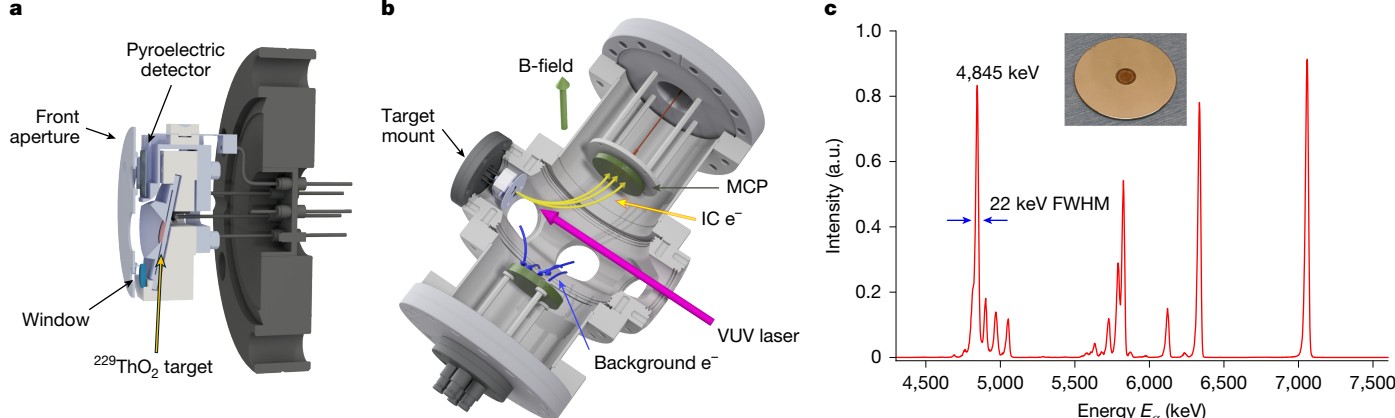

**Fig. 1 | CEMS target set-up and characterization. a**, Cutaway rendering of the $^{229}ThO_2$ target mount. Arrows denote front aperture, window, target and pyroelectric detector. **b**, Rendering of the spectroscopy chamber. Magenta arrow, direction of VUV laser propagation. Yellow arrows, IC electron trajectories from target to detection MCP. Blue arrows, background photoelectrons generated from VUV scatter diverted to the secondary electrode. Green arrow, direction of static B-field used to guide IC electrons. **c**, α-spectrum of the $^{229}ThO_2$ target. Inset, photograph of $^{229}ThO_2$ target used in this study. The peak indicated at 4,845 keV corresponds to the dominant α-decay mode of $^{229}Th$. The other large peaks correspond to the α-decays of daughter nuclei. The $^{229}Th$ peak had a full width at half maximum (FWHM) of about 22 keV, consistent with energy loss through an approximately 10-nm sample, as estimated with SRIM[56]. a.u., arbitrary units.

As well as providing a means to study low-bandgap materials containing $^{229}Th$, CEMS could also allow marked improvements in nuclear clock stability, as the IC decay rate is roughly $10^8$ times faster than the radiative decay rate, enabling a much faster clock interrogation rate. This rapid interrogation reduces the stability demands on the local oscillator while providing a projected clock instability as low as about $10^{-18}$ at 1 s. Further, a CEMS-based nuclear clock could operate by simply reading out the CEMS photocurrent, providing a means to greatly simplify and miniaturize future nuclear clocks[32].

Here we report the first demonstration of laser-based CEMS of any nucleus. Specifically, a VUV laser system is used to excite $^{229}Th$ nuclei in a thin sample of thoria ($ThO_2$). $^{229}ThO_2$ was chosen as the first host as it has a low bandgap (about 6 eV (ref. 33)) and is readily available as a stoichiometric Th compound. Conversion electrons from the $^{229}ThO_2$ are detected as a function of laser excitation energy providing the first laser-based CEMS nuclear spectrum, which determines the $^{229}Th$ nuclear transition frequency as 2,020,407.5(2)$_{stat}$(30)$_{sys}$ GHz, consistent with previously reported values of the transition[14–16,19] and whose width is consistent with the laser linewidth of our system reported in previous studies[15,19]. Further, the IC lifetime is measured as approximately 10 μs, which is consistent with that measured in $^{229}Th$ implantation studies[34] and theoretical calculations of the IC lifetime described below. In what follows, we describe the $^{229}ThO_2$ target, experimental apparatus and the spectroscopic protocol, before presenting the results of the CEMS spectroscopy. Next, we present calculations of the isomer shift and IC lifetime in $^{229}ThO_2$ using a theoretical approach that combines solid-state density functional theory (DFT) and relativistic atomic many-body perturbation theory methods. Following these results, we use the measured lifetime of the IC decay of the isomeric state in $^{229}ThO_2$ to evaluate the ideal performance of a $^{229}ThO_2$ conversion electron clock and find its projected instability to be about $2 × 10^{-18}$ at 1 s.

## $^{229}ThO_2$ target and apparatus

The CEMS target was constructed by electrodepositing $^{229}ThO_2$ onto a stainless steel disc (see Fig. 1 for an image of the target, procured from Eckert & Ziegler Analytics, Inc.). Although other oxide and hydroxide compounds of Th may be present in the target, this paper assumes that the predominant chemical form of $^{229}Th$ in the target is $^{229}ThO_2$, based on the method of target preparation (see Methods for details). The diameter of the $^{229}ThO_2$ is about 5 mm and its thickness is estimated from the width of the α-particle spectrum as approximately 10 nm (Fig. 1c),

which is consistent with a measured total activity of about 6.3 kBq. The VUV absorption of $ThO_2$ is estimated as $α ≈ 0.1$ nm$^{-1}$ (ref. 35), implying that $^{229}Th$ nuclei within a depth of about 10 nm can be efficiently excited by the laser. Also, the inelastic mean free path of electrons in solids at the eV energy scale is typically about 10 nm (ref. 36), meaning that an IC electron generated at greater depths probably scatters before extraction from the surface. As such, further $^{229}ThO_2$ thickness primarily contributes to the background in the form of electrons produced through radioactive decays[37] and thus a thickness of about 10 nm is probably optimal for CEMS.

CEMS spectroscopy is recorded by directing a tunable VUV laser onto the $^{229}ThO_2$ target. Briefly, VUV radiation was produced by means of resonance-enhanced four-wave mixing of two pulsed dye lasers in Xe gas. The frequency of the first pulsed dye laser, $ω_u$, was locked to the $5p^6\,^1S_0 → 5p^5\left(^2P^°_{3/2}\right)6p^2\,[1/2]_0$ two-photon transition of Xe at about 249.63 nm. The frequency of the second pulsed dye laser, $ω_v$, was scanned to produce VUV radiation in the Xe cell given by the difference mixing relation $ω = 2ω_u − ω_v$. All three laser beams then impinge off-axis with respect to a $MgF_2$ lens, whose chromatic dispersion is used with downstream pinholes to spatially filter the VUV beam and pass it towards the $^{229}ThO_2$ spectroscopy chamber. The laser system delivers 30 pulses per second to the target with a typical VUV pulse energy of about 6–8 μJ per pulse (see refs. 15,38 for details). For these experiments, $N_2$ gas is added to the Xe cell to quench an amplified spontaneous emission (ASE) process at 147 nm (see Methods for details), which occurs on a timescale similar to IC decay and otherwise contributes to the photoelectron background.

The $^{229}ThO_2$ target is held in a custom mount (see Fig. 1a) that provides the ability to electrically bias the sample, as well as VUV laser monitoring capabilities. Biasing the target is necessary as the VUV laser causes a prompt burst of photoelectrons when it impinges on the $^{229}ThO_2$ target, which will overwhelm the electron detector. These initial photoelectrons are suppressed for approximately the first 100 ns after the laser pulse by positively biasing the $^{229}ThO_2$ target to 135 V, before the bias voltage is changed to −405 V to aid the extraction of IC electrons from the surface; at all times, the front aperture of the mount is held at 100 V. The mount also houses a fused silica target that fluoresces under VUV illumination aiding alignment and a pyroelectric crystal to provide in situ monitoring of the VUV laser pulse energy. The target chamber is mounted on a combined lever arm–stepper motor system that allows it to be raised and lowered, causing the VUV laser to illuminate either the $^{229}ThO_2$ target or the laser energy monitor. To limit hydrocarbon

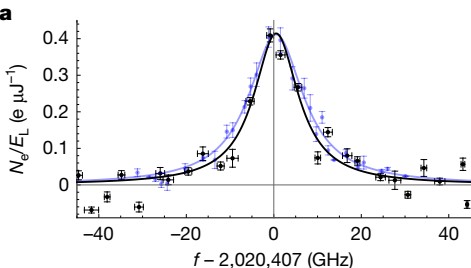

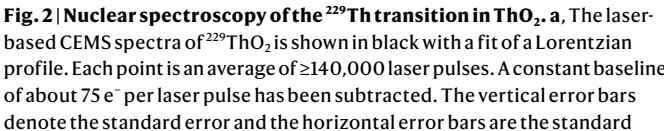

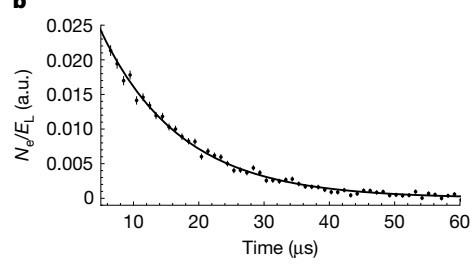

**Fig. 2 | Nuclear spectroscopy of the $^{229}$Th transition in ThO$_2$. a**, The laser-based CEMS spectra of $^{229}$ThO$_2$ is shown in black with a fit of a Lorentzian profile. Each point is an average of ≥140,000 laser pulses. A constant baseline of about 75 e$^-$ per laser pulse has been subtracted. The vertical error bars denote the standard error and the horizontal error bars are the standard

deviation of the laser frequency for the points in the bin. For comparison, the radiative decay spectrum of $^{229}$Th:LiSrALF$_6$, taken from ref. 15, is shown in blue. **b**, The IC decay rate as a function of time is shown in black, alongside a fit of a decaying exponential with a background, leading to an IC lifetime of 12.3(3) µs. a.u., arbitrary units.

build-up on the target and to limit scattering of the IC electrons owing to background gases, the target chamber was ozone plasma cleaned before baking and operated at a pressure of about $10^{-7}$ Pa.

In principle, IC electron detection can be accomplished by simply positively biasing, for example, a multichannel plate (MCP) detector held near the $^{229}$ThO$_2$. However, it was found that scattered light from the VUV laser generated a large afterglow of photoelectrons originating at positions dispersed throughout the chamber, which lasted tens of microseconds—presumably because of the fluorescence of components within the beamline and chamber. To overcome this background, a combination of electric and magnetic fields was used that focused electrons originating from the $^{229}$ThO$_2$ target onto a detection MCP, while diverting electrons generated elsewhere to another region of the chamber (see Methods). Finally, as an extra layer of protection for the detection MCP, its front plate voltage was controlled so that it had no gain during the initial burst of photoelectrons from the target.

## Spectroscopy

Using this apparatus, CEMS was performed by recording the number of detected electrons in the window between 6 and 40 µs after each laser pulse as a function of VUV laser wavelength; this window was chosen on the basis of previous theoretical estimates[23] and observations of IC decay[39] following $^{233}$U decay. The electron signal, normalized by the laser pulse energy, is shown in Fig. 2a as a function of frequency for a region roughly 100 GHz wide and centred on the $^{229}$Th nuclear transition energy. These data were recorded by first measuring the pulse energy and then detecting the total number of electrons emitted after 36,000 pulses (20 min) before measuring the pulse energy again. This process was repeated to record a laser CEMS spectrum. The data in Fig. 2a are the average of four spectra, with the vertical error bars representing the standard error. Owing to differences in system alignment between campaigns, the photoelectron background varied slightly between different spectra. Therefore, a Lorentzian with a background fit was used to determine and subtract the photoelectron background before the data are combined. Further, because the laser frequency is not perfectly controlled, these spectra are binned in frequency and the horizontal error bars are the standard deviation of the frequency. Each data point represents the average of at least 140,000 laser pulses.

A nonlinear least-squares fit of a Lorentzian to the spectrum yields a central value of 2,020,407.5(2)$_{stat}$(30)$_{sys}$ GHz and full width at half maximum linewidth of 12.4(4) GHz, both consistent with our observations in LiSrAlF$_6$ (ref. 15) and ThF$_4$ (ref. 19). Here statistical error is the 68% confidence interval and the systematic uncertainty is predominantly because of the accuracy of the wavemeter and probably conservative. On resonance, an average of 0.41(5) e µJ$^{-1}$ per laser pulse is detected. The expected signal can be estimated as $\eta_e\eta_c N_e \times e^{-t_i/\tau_{IC}}$, in which $\eta_e$ is the extraction of efficiency of IC electrons from the sample,

$\eta_c$ is the collection efficiency of electrons emitted from the sample, $N_e$ is the number of excited $^{229}$Th nuclei and $t_i$ is the time that electron counting begins after the laser pulse. On the basis of calibrations of the apparatus (see Methods), the expected signal is found as $0.15^{+0.25}_{-0.09}$ e$^-$ µJ$^{-1}$, in reasonable agreement with the recorded spectrum.

Using the same system, the lifetime of the IC decay is measured by comparing the time-binned electron counts collected on resonance in the first 190 µs after laser excitation. Specifically, the average number of time-binned electron counts per µJ of laser energy is obtained for both the on-resonant and off-resonant (about 100 GHz detuning) illumination periods. The average off-resonance counts are then subtracted from the average on-resonance counts to remove any photoelectron background and the result is plotted in Fig. 2b. A background-free nonlinear least-squares fit of a decaying exponential reveals an IC lifetime of 12.3(3) µs, as shown in Fig. 2b. As discussed later, the IC decay rate is sensitive to the local chemical environment, so a range of decay rates is possible and this lifetime should be interpreted as an estimate.

## Calculation of isomer shifts and IC lifetime

The measured isomer energy is in agreement with the earlier spectroscopic experiments in three wide-bandgap hosts[14,15,19]. Theoretically, this is supported by the calculations[40] of isomer shifts for a large variety of $^{229}$Th solid-state hosts and by our $^{229}$ThO$_2$-specific calculations presented in Methods. For example, we expect that the isomer shift between bulk $^{229}$ThO$_2$ and $^{229}$Th:LiSrAlF$_6$ is on the order of 100 MHz, well below the approximately 3 GHz reported experimental accuracy. Our calculations predict the nuclear transition frequency for bulk $^{229}$ThO$_2$ to be 2,020,407,338(70) MHz.

In the IC process, the energy of the excited nucleus is transferred to the electrons. In a crystalline solid, that means promoting a valence band electron $|v\mathbf{k}\sigma_v\rangle$ into a conduction band state $|c\mathbf{k}\sigma_c\rangle$ and creating a hole in the valence band (Fig. 3a). The process is mediated by the hyperfine interaction (HFI) $W$, which, for periodic lattices, conserves the electronic crystal momentum $\mathbf{k}$ but can connect electronic states of different spin projections $\sigma_{v,c}$. We derived the rate for the described IC process (see Methods) as

$$\Gamma_{IC} = \frac{2\pi}{\hbar} \frac{1}{2I_e+1} \sum_{M_g M_e} \sum_{\sigma_c \sigma_v} \overline{|W^{ge}_{c\sigma_c v\sigma_v}(\mathbf{k})|^2} G_{cv}(\hbar\omega_{nuc}). \quad (1)$$

Here $|e\rangle$ and $|g\rangle$ are the isomer and ground nuclear states with spins $I_{e,g}$ and magnetic quantum numbers $M_{e,g}$. The HFI $W$ connects the two nuclear states and the valence and conduction band electronic states. $G_{cv}(\hbar\omega_{nuc})$ is the conventional[41] joint density of states (JDOS) at the nuclear transition energy $\hbar\omega_{nuc}$. The bar denotes averaging over a surface of constant energy in $\mathbf{k}$-space.

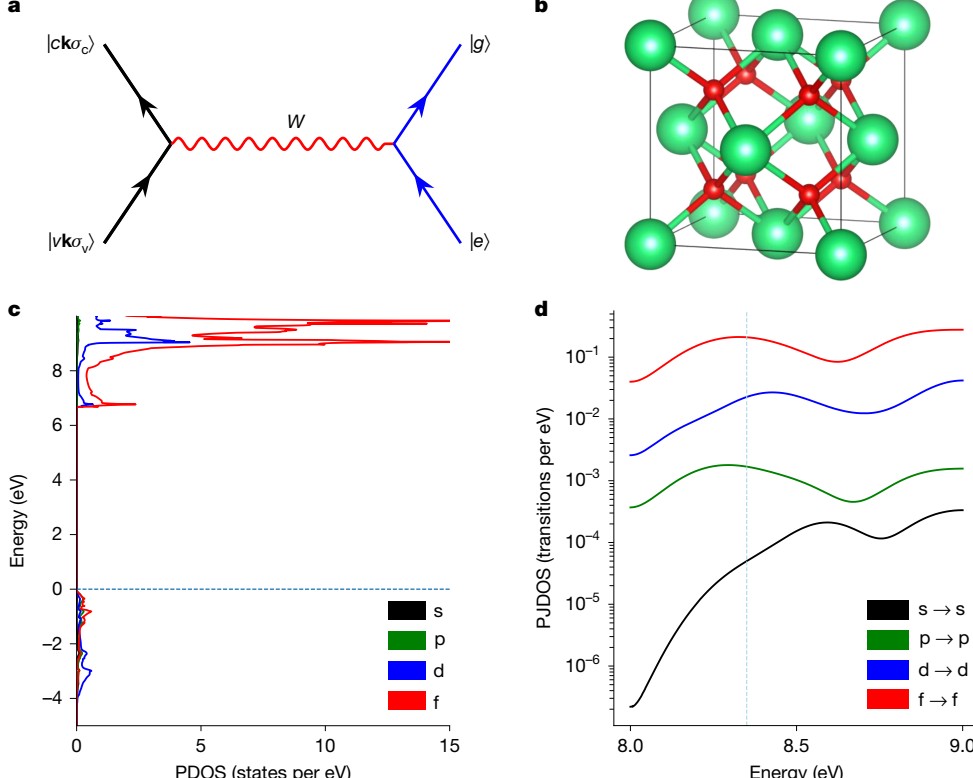

**Fig. 3 | Calculation of the IC rate in ThO$_2$. a**, Feynman diagram of the IC process. Single black strokes represent the electron lines and the double blue ones denote the nuclear lines. The interaction is mediated by a virtual photon exchange, corresponding to the HFI $W$. The crystal momentum **k** is conserved in this process. **b**, Conventional unit cell of ThO$_2$, with Th atoms in green and O atoms in red. **c**, Th PDOS computed with $G_0W_0$. The horizontal dashed line denotes the Fermi level. **d**, Th PJDOS computed with $G_0W_0$. The vertical dashed line denotes the nuclear isomeric energy.

The same JDOS appears in the theory of electromagnetic interband absorption[41] and is clearly a highly material-dependent quantity. Therefore, to understand the physics of the IC process, and in particular its rate, we calculated the properties of $^{229}$ThO$_2$ with periodic electronic structure theory ($GW$ and DFT). The calculations used the $G_0W_0$ approximation for the electronic self-energy and the Bethe–Salpeter equation (BSE) for optical properties (see Methods for details). The fundamental bandgap computed with $G_0W_0$ is 6.20 eV, in reasonable agreement with experimental measurements of 5.9 eV (refs. 42,43) (calculation details provided in Methods). The absorption spectrum, computed with $G_0W_0$ + BSE, agrees with the measurements in ref. 35, supporting our estimation of $\alpha \approx 0.1$ nm$^{-1}$.

ThO$_2$ crystallizes in the fluorite structure (space group $Fm\bar{3}m$, #225), in which each Th atom is coordinated by eight oxygen atoms in a cubic arrangement (Fig. 3b). The Th$^{4+}$–O$^{2-}$ bonding is predominantly ionic, with a modest covalent admixture arising from overlap of Th 6d/5f and O 2p orbitals. That Th is in the +4 oxidation state may be verified from the projected density of states (PDOS) on Th, in which the 5f and 6d components in the valence band are nearly empty. By contrast, the Th 5f and 6d valence orbitals form large peaks in the conduction band, indicating the transfer of an electron from an O$^{2-}$ anion to form Th$^{3+}$. Both the valence and conduction bands contain small contributions from Th-projected p orbitals. As it turns out, the IC process primarily involves transferring electrons between these p-orbital components.

In computing the HFI matrix element in equation (1), it is convenient to project the valence and conduction band states $|v\mathbf{k}\sigma_v\rangle$ and $|c\mathbf{k}\sigma_c\rangle$ onto the basis of Th atomic orbitals. The matrix element $W_{c\sigma_c v\sigma_v}^{ge}(\mathbf{k})$ may then be expressed as a sum over different HFI matrix elements connecting atomic orbitals of definite angular momenta, weighted by the expansion coefficients of $|v\mathbf{k}\sigma_v\rangle$ and $|c\mathbf{k}\sigma_c\rangle$. To the lowest order, cross terms arising when squaring the expansion of $W_{c\sigma_c v\sigma_v}^{ge}(\mathbf{k})$ are neglected.

This allows the expression of the IC rate in terms of the nuclear ground state HFI constants $A$ and the 'projected joint density of states' (PJDOS) in Th. The PJDOS is a JDOS weighted by the projections of the crystal electronic states onto the thorium atomic orbitals that allows us to describe the character of the delocalized crystal orbitals close to the thorium nucleus (see Methods). The HFI $A$ constants for Th$^{3+}$ are known from experiments and high-precision relativistic atomic-structure calculations[12]. The PJDOS of transitions with the largest contributions to the IC rate are shown in Fig. 3c. Using these values, we arrive at an estimate for the IC rate of $\Gamma_{IC} \approx 1.3 \times 10^4$ s$^{-1}$, corresponding to an IC lifetime of about 80 µs.

Sources of the disparity between the measured and theoretical lifetimes could be because of cross-order and higher-order expansion terms neglected in our evaluation of the rate (equation (1)) and errors in the local projections of the plane-wave orbitals done by VASP[44]. For example, adding the cross terms coherently may increase the IC rate and thus reduce the estimated IC lifetime to approximately 30 µs. Relativistic contraction approximately $(Z\alpha)^2$ may increase the non-relativistic PDOS computed with VASP as much as 40%, so the increase in the IC rate may be up to a factor of $1.4^2 \approx 2$, further reducing the estimated IC lifetime to about 15 µs. See Methods for more details. Small changes to the valence and conduction states expected with the addition of spin–orbit coupling in solid-state calculations and deviations of the sample from bulk $^{229}$ThO$_2$ owing to both surface and self-radiation damage effects may further affect the IC rate. Compared with the pristine $^{229}$ThO$_2$, the IC rate for $^{229}$Th adjacent to point defects can be enhanced by the electric quadrupole HFI contribution[45] owing to symmetry breaking. Evaluating these uncertainties requires a much more comprehensive study that will be carried out in future work. Here we simply note that the order-of-magnitude experiment–theory agreement supports the physical interpretation that observed

IC decay results from the nucleus relaxing by means of transferring an electron from an oxide anion into the Th 6p component of the conduction band.

## Projected clock performance

As well as its use as a new chemical probe, laser-based CEMS may allow the construction of a new type of nuclear clock with several advantages over crystal-based clocks[46]. Chief among these advantages are a greatly reduced clock interrogation time, as the IC decay rate is about $10^8$ times faster than radiative decay, and the potential for clock readout by simply monitoring the current leaving the target, which could facilitate substantial miniaturization.

For such a clock, assuming that $^{229}ThO_2$ is produced from $^{16}O$, the largest sources of instability would be broadening owing to magnetic dipole interactions between the $^{229}Th$ nuclei and lifetime broadening. Given that the Th–Th distance in $ThO_2$ is 3.96 Å (ref. 47), the expected Zeeman broadening owing to neighbouring $^{229}Th$ nuclei is about 10 Hz. The next largest sources of broadening would be the varying intrinsic isomeric shift and second-order Doppler shift arising from temperature gradients across the $ThO_2$ sample. Drawing estimates from other Mössbauer experiments, typical intrinsic isomeric shifts range from 0.1 to 5.0 kHz $K^{-1}$ (refs. 48–51), as supported by recent measurements of the intrinsic isomeric shift in $^{229}Th:CaF_2$ at 0.4 kHz $K^{-1}$ (ref. 11). Within the Debye model, the second-order Doppler shift can be estimated to be $\lesssim 1$ Hz $K^{-1}$ at 4 K, using the lowest reported value of 236 K for the Debye temperature of $ThO_2$ (ref. 52). With temperature gradients stabilized across the sample to $\leq 0.1$ mK, the intrinsic isomeric and second-order Doppler shifts result in a broadening of roughly 0.5 Hz. Thus, the primary source of instability is the lifetime broadening of $\Gamma \approx 2\pi \times 16$ kHz.

Assuming that a polished face of an approximately 10-nm-thick single crystal is prepared and realizes an electron extraction efficiency $\eta_e \approx 0.5$ (ref. 36) and that the photoelectron background is eliminated by material purification and baffling, a 100-µW laser leads to a projected clock instability of about $2 \times 10^{-18}$ at 1 s averaging (see Methods for details). Notably, it is possible that the electric current from the IC process could also be used as a form of readout, as it would generate a current of about 300 nA, providing a simple means for clock operation and locking[32]. It may also be possible to further enhance this performance by purposely exciting defect or conduction band electrons to realize an 'on-demand' quench of the nuclear excitation[17,18].

## Discussion and outlook

With this first demonstration of laser-based CEMS, an entirely new class of materials is now compatible with laser nuclear spectroscopy. By implanting $^{229}Th$ into low-bandgap materials and observing both the lifetime of the IC decay channel and the isomer shift of the transition, detailed information can be gathered about the local phononic, electronic and nuclear structure. This in turn allows the isomeric transition to serve as a sensor of strain and temperature in the solid[11,3]. Further, at the energy scale of the IC electrons, their inelastic mean free path is a sensitive function of their energy relative to the Fermi level[36]. In the future, it may be possible to relate the efficiency with which IC electrons emerge from a surface with both the surface quality and the band structure local to the implanted $^{229}Th$. Looking further forward, off-resonant excitation of $^{229}Th$ nuclei through a combined nuclear and phononic transition in the solid could enable laser-based nuclear resonance vibrational spectroscopy[53,54]. This could allow measurement of the local phonon density of states with sub-µeV precision, considerably beyond what is possible with conventional Mössbauer spectroscopy[54].

Given our previous work in $^{229}ThF_4$ films[19], it can be concluded that, as the $^{229}Th$ is converted from the oxide to the fluoride state, the $^{229}Th$ isomer will convert from decaying by IC to decaying by the emission of a VUV photon. Thus, the relative ratio of radiative decay to conversion electron decay following laser excitation could provide a new set of methods to characterize the chemical state of Th compounds, perhaps aiding studies of nuclear power generation[55].

Furthermore, laser-based CEMS provides a new platform to realize a solid-state Th clock that benefits from the inherent ease of a production of stoichiometric Th compounds, a $10^8$ reduction in clock interrogation cycle and the potential for a current-based readout, allowing simplification and miniaturization of future nuclear clocks.

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

## Methods

### Expected IC signal

The incident VUV flux is attenuated, so that it follows $\varphi_0 e^{-\alpha z}$, in which $\alpha$ is the VUV attenuation coefficient. The number of excited $^{229}$Th nuclei in a target of length $L$ should go as[57]

$$N_e \approx \frac{4}{6} \frac{\lambda^2}{2\pi} \frac{n_{Th}}{\Gamma_L} \frac{T|\tilde{n}|^2}{Re[\tilde{n}]} \frac{\varphi_0(1-e^{-\alpha L})}{\alpha} \times \frac{1}{1+4\left(\frac{\delta}{\Gamma_L}\right)^2} \times \left(\frac{t_e}{\tau_{rad}}\right), \quad (2)$$

in which $\lambda$ is the vacuum transition wavelength, $n_{Th}$ is the density of $^{229}$Th in the $^{229}$ThO$_2$ target, $\Gamma_{rad} = 1/\tau_{rad}$ is the vacuum radiative decay rate, $\Gamma_L$ is the VUV laser bandwidth, $\Gamma_{IC} = 1/\tau_{IC}$ is the IC decay rate, $\delta$ is the laser detuning, $\varphi_0$ is the incident laser photon flux, $L$ is the target thickness, $T$ is the transmission of the VUV laser into the target and $\tilde{n} = n - i\kappa$ is the complex index of refraction with $\kappa = \lambda\alpha/4\pi$. The target was produced using a 0.75:0.25 $^{229}$Th:$^{232}$Th isotope mix from Oak Ridge National Laboratory, leading to an effective $^{229}$Th density of $n_{Th} = 0.75 \times 2.28 \times 10^{22}$ cm$^{-3}$ (ref. 47).

From the number of excited $^{229}$Th nuclei, the number of detected IC electrons is given by

$$N_{det} = \eta_e \eta_c N_e \times e^{-t_i/\tau_{IC}} \quad (3)$$

in which $\eta_e$ is the extraction efficiency from the $^{229}$ThO$_2$ target, $\eta_c$ is the collection efficiency and $t_i$ is the start of the acquisition time window. The extraction efficiency represents the probability that an IC electron is able to leave the $^{229}$ThO$_2$ target and combines many physical processes, such as whether the electron is promoted high enough into the conduction band to overcome the ionization energy barrier (either by being in a state above the ionization energy or tunnelling), whether the electron inelastically scatters, whether surface conditions are favourable and so on. Owing to all of these confounding factors, theoretical calculation of $\eta_e$ is difficult and it must be estimated from experiments. To do this, we make the assumption that the probability for a photoelectron promoted by a VUV photon to leave the target is the same as an IC electron promoted by an excited $^{229}$Th nucleus. By making this assumption, we are able to use the efficiency with which photoelectrons are generated by our VUV laser system to obtain an estimate of $T\eta_e$. The collection efficiency $\eta_c$ is simply the efficiency with which electrons that leave the target are detected on the MCP given the voltage biasing and magnetic field configuration used in the experiment. This can be readily determined by measuring the ratio of the number of photoelectrons collected on the MCP front plate relative to the number of photoelectrons leaving the target.

Using the values listed in Extended Data Table 1, the expected on-resonance IC signal was estimated to be $N_{det}/(\hbar\omega_0\varphi_0 t_e) = 0.15^{+0.25}_{-0.09}$ e$^-$ µJ$^{-1}$ per shot.

### Quenching of ASE in Xe four-wave mixing

The four-wave mixing process can produce a beam of ASE along the beam axis, complicating detection. The origin of the ASE is resonant excitation by the 249.6-nm laser of the $5p^6\ ^1S_0 \rightarrow 5p^5\left(^2P^o_{3/2}\right)6p^2\ [1/2]_0$ transition. Any Xe population not participating in the four-wave mixing process will be left in the excited two-photon state. This $5p^5\left(^2P^o_{3/2}\right)6p^2\ [1/2]_0$ (abbreviated as $6p^2\ [1/2]_0$) state will then decay to the $5p^5\left(^2P^o_{3/2}\right)6s^2\ [3/2]^o_1$ (abbreviated as $6s^2\ [3/2]^o_1$) state by 828-nm emission in about 30 ns and the $6s^2\ [3/2]^o_1$ state will decay to the $^1S_0$ ground state by emission of a 147-nm photon in about 3.7 ns. If the Xe pressure is in the several hundred Pa range, as it is for efficient four-wave mixing, there are enough nearby Xe atoms that they may reabsorb these 828-nm and 147-nm photons. Effectively, this leads to radiation trapping, which extends the effective fluorescence lifetime of the Xe spontaneous emission[58]. At the same time, the 828-nm/147-nm spontaneous emission will experience gain as it stimulates Xe in the excited $6p^2\ [1/2]_0$ and $6s^2\ [3/2]^o_1$ to emit, yielding ASE. This gain will be

highly directional, as the excited Xe will essentially lie in a column defined by the propagation of the 249.6-nm pulse of the pump laser. This interplay between radiation trapping and ASE will then yield bidirectional emission from the Xe cell along the pumping axis, which will have a timescale much longer than the spontaneous emission lifetime of the excited states[59].

To mitigate this effect, we introduce N$_2$ gas in the Xe, which quenches the excited state Xe population through collision.

### Simulation of electron trajectories

SIMION[60] simulations were carried out to determine a combination of voltage biasing and static magnetic field that would guide IC electrons to our detector while diverting background photoelectrons to a secondary electrode. The voltage biasing chosen during the IC observation period was to have the target at −405 V, the front aperture of the target mount at +135 V, the front of the detection MCP at +140 V and the secondary electrode at +2,500 V. The magnetic field was set to about 3–5 G. As can be seen in Extended Data Fig. 1a, under the voltage biasing described, there is a 'saddle' in the potential through which the IC electrons that have been accelerated by our target mount system can be made to travel through by the magnetic field. Meanwhile, as can be seen in Extended Data Fig. 1b, electrons that are generated randomly throughout the chamber with about 8 eV of kinetic energy either crash into the chamber wall or fall into the sacrificial electrode.

### Preparation of the target

The target used in this study was prepared in an electrodeposition buffer solution of NaOH and sulfuric acid. $^{229}$Th material was deposited onto the stainless steel disc until target activity was reached, at which a quench was performed with ammonium hydroxide. The disc was then heated overnight at 200–300 °C in ambient atmosphere.

Although the electrodeposited target is probably a mixture of Th hydroxides, oxides and metal impurities, the use of low-impurity (>99.99% grade) chemicals during electrodeposition and heat treatment in air lead us to assume that it is predominantly Th in an oxidized state. As the most stable oxide form, we assume that the target is predominately $^{229}$ThO$_2$.

### IC rate derivation

We consider the IC processes in an insulator, when the bandgap $\Delta$ is smaller than the nuclear isomer energy $\omega_{nuc}$. In the IC process, the nuclear excitation is transferred to the electrons, spawning a particle–hole pair, with a valence band electron promoted into the conduction band, leaving behind a hole in the valence band; see Fig. 3a.

The $^{229}$Th nuclear subsystem is modelled as two distinct energy levels: the ground state $|g\rangle$ with nuclear spin $I_g = 5/2$ and the excited (isomeric) state $|e\rangle$ with $I_e = 3/2$, separated by the energy gap $\omega_{nuc}$.

The IC process is mediated by the HFI. In a crystal, containing $\mathcal{N}$ $^{229}$Th nuclei with one $^{229}$Th per unit cell, HFI reads

$$W(\mathbf{r}) = \sum_{\nu=1}^{\mathcal{N}} \mathcal{M}(\mathbf{R}_\nu) \cdot \mathcal{T}(\mathbf{r} - \mathbf{R}_\nu), \quad (4)$$

in which we sum over unit cells, $\mathcal{M}(\mathbf{R}_\nu)$ is a $^{229}$Th nuclear magnetic moment operator and the $^{229}$Th-centred $\mathcal{T}$ is a rank-1 tensor acting on the electronic degrees of freedom. The nuclear electric-quadrupolar and higher-rank HFI interactions can be added in a similar fashion. For $^{229}$ThO$_2$, owing to the cubic symmetry, the electric-quadrupole contribution vanishes.

To compute the IC rate, we use Fermi's golden rule. The initial state of the electron subsystem is a fully occupied valence band $|\tilde{O}\rangle$ and the final state is the particle–hole excitation $a^\dagger_{c\mathbf{k}\sigma_c} a_{v\mathbf{q}\sigma_v}|\tilde{O}\rangle$ with energy $\varepsilon_{c\mathbf{k}} - \varepsilon_{v\mathbf{q}}$, in which $\varepsilon_{c\mathbf{k}}$ and $\varepsilon_{v\mathbf{q}}$ are the band functions of the conduction (c) and the valence (v) bands. Then, the final state quantum numbers are spanned by crystal momenta $\mathbf{k}$, $\mathbf{q}$ and two electron spin projections. We also sum

over the nuclear ground state magnetic quantum numbers $M_g$ and average over the nuclear isomer state magnetic quantum numbers $M_e$,

$$\Gamma_{IC}^{(\mathcal{N})} = \frac{2\pi}{\hbar} \frac{1}{2I_e+1} \sum_{M_g M_e} \sum_{\sigma_c \sigma_v} \int_{BZ} d^3k \int_{BZ} d^3q |\langle c\mathbf{k}\sigma_c|W^{ge}|v\mathbf{q}\sigma_v\rangle|^2$$
$$\delta(\varepsilon_{c\mathbf{k}} - \varepsilon_{v\mathbf{q}} - \omega_{nuc}). \tag{5}$$

The integrations are carried over the Brillouin zone (BZ). We use the conventions from ref. 41 in our derivation.

The HFI (equation (4)) is cell-periodic and its matrix element $\langle c\mathbf{k}\sigma_c|W^{ge}|v\mathbf{q}\sigma_v\rangle$ between Bloch functions can be reduced from an integration over the entire crystal to an integration over a single unit cell. The result reads

$$\langle c\mathbf{k}\sigma_c|W^{ge}|v\mathbf{q}\sigma_v\rangle = \delta(\mathbf{k}-\mathbf{q})W_{c\sigma_c v\sigma_v}^{ge}(\mathbf{k}) \tag{6}$$

with

$$W_{c\sigma_c v\sigma_v}^{ge}(\mathbf{k}) \equiv \frac{(2\pi)^3}{\Omega}\langle g|\mathcal{M}|e\rangle \cdot \int_\Omega d^3r u_{c\mathbf{k}}^*(\mathbf{r})\chi_{\sigma_c}^\dagger \mathcal{T}(\mathbf{r})\chi_{\sigma_v} u_{v\mathbf{k}}(\mathbf{r}). \tag{7}$$

Here $\Omega$ is the unit cell volume, $u_{...}(\mathbf{r})$ are cell-periodic envelopes of Bloch functions[41] and $\chi$ are the conventional electron spinors. Note that the cell-periodicity imposes conservation of crystal momentum, $\mathbf{k} = \mathbf{q}$.

Now, with this matrix element, we evaluate the IC rate.

$$\Gamma_{IC}^{(\mathcal{N})} = \frac{2\pi}{\hbar} \frac{1}{2I_e+1} \sum_{M_g M_e} \sum_{\sigma_c \sigma_v} \int d^3k \int d^3q |W_{c\sigma_c v\sigma_v}^{ge}(\mathbf{k})|^2$$
$$\delta(\mathbf{k}-\mathbf{q})\delta(\mathbf{k}-\mathbf{q})\delta(\varepsilon_{c\mathbf{k}} - \varepsilon_{v\mathbf{q}} - \omega_{nuc}). \tag{8}$$

There is a product of two identical delta functions. $\mathbf{q}$ is replaced by $\mathbf{k}$ while integrating over $\mathbf{q}$, but we encounter $\delta(0) = \lim_{\Delta \mathbf{q}\to 0}\delta(\Delta \mathbf{q})$. Using the following identity for each component of $\Delta \mathbf{q}$:

$$\lim_{\Delta q_x \to 0}\delta(\Delta q_x) = \lim_{L_x \to \infty}\cos\frac{1}{2\pi}\int_{-L_x/2}^{L_x/2}e^{i\Delta q_x x}dx = \frac{L_x}{2\pi},$$

in which $L_x$ is the crystal size in the $x$-direction, we can show that

$$\lim_{\Delta \mathbf{q}\to 0}\delta(\Delta \mathbf{q}) = \frac{V_{xtal}}{(2\pi)^3},$$

in which the crystal volume $V_{xtal} = L_x L_y L_z$. This is similar to the formal time-domain limit in deriving Fermi's golden rule; see, for example, p. 72 of ref. 61.

Thereby, the IC rate per crystal volume

$$\Gamma_{IC}^{(\mathcal{N})}/V_{xtal} = \frac{2\pi}{\hbar} \frac{1}{(2\pi)^3} \frac{1}{2I_e+1} \sum_{M_g M_e} \sum_{\sigma_c \sigma_v} \int_{BZ} d^3k |W_{c\sigma_c v\sigma_v}^{ge}(\mathbf{k})|^2$$
$$\delta(\varepsilon_{c\mathbf{k}} - \varepsilon_{v\mathbf{k}} - \hbar\omega_{nuc}) \tag{9}$$

The assumption made during the derivation was that all $\mathcal{N}$ $^{229}$Th nuclei were initially in the excited (isomer) state. Consider an ensemble of quantum emitters, with $N(t)$ being the number of emitters in the excited state at time $t$, with a single emitter decay rate $\gamma$ and

$$dN/dt = -\gamma N(t) \Rightarrow N(t) = N(0)\exp{-\gamma t}$$
$$\Rightarrow \text{ ensemble decay rate } \Gamma$$
$$= -dN/dt \tag{10}$$
$$= \gamma N(0)\exp{-\gamma t}.$$

We see that the ensemble decay rate $\Gamma$ at $t = 0$ is $\gamma N(0)$. The experimentally relevant excited state population, however, decays as $\exp{-\gamma t}$.

On the basis of this discussion, we define the experimentally measured IC decay rate

$$\Gamma_{IC} = \Gamma_{IC}^{(\mathcal{N})}/\mathcal{N} = \Omega \Gamma_{IC}^{(\mathcal{N})}/V_{xtal}. \tag{11}$$

Following the derivation of interband electromagnetic absorption rates[41], we define a surface $S_E$ of constant energy in $\mathbf{k}$-space through an implicit relation $\varepsilon_{c\mathbf{k}} - \varepsilon_{v\mathbf{k}} = \hbar\omega_{nuc}$. If the matrix element remains reasonably constant on $S_E$, the rate simplifies to

$$\Gamma_{IC} = \frac{1}{\tau_{IC}} = \frac{2\pi}{\hbar} \frac{1}{2I_e+1} \sum_{M_g M_e} \sum_{\sigma_c \sigma_v} \overline{|W_{c\sigma_c v\sigma_v}^{ge}(\mathbf{k})|^2} G_{cv}(\hbar\omega_{nuc}). \tag{12}$$

with the JDOS

$$G_{cv}(E) = \frac{\Omega}{(2\pi)^3}\int_{BZ} d^3k\delta(\varepsilon_{c\mathbf{k}} - \varepsilon_{v\mathbf{k}} - E) = \frac{\Omega}{(2\pi)^3}\int \frac{dS_E}{|\nabla_\mathbf{k}(\varepsilon_{c\mathbf{k}} - \varepsilon_{v\mathbf{k}})_E|}. \tag{13}$$

With this simplification, $\overline{|W_{c\sigma_c v\sigma_v}^{ge}(\mathbf{k})|^2}$ has the meaning of $|W_{c\sigma_c v\sigma_v}^{ge}(\mathbf{k})|^2$ averaged over the surface of constant energy $S_E$.

This concludes the derivation of the IC rate (equation (1)).

Also worthy of mention, note that the electromagnetic absorption coefficient at laser frequency $\omega_{nuc}$ is proportional to the same JDOS $G_{cv}(\hbar\omega_{nuc})$. This can be used to back out the JDOS value from the laser absorption measurements at a frequency slightly detuned away from $\omega_{nuc}$.

To proceed with the computation of the hyperfine matrix element, we expand the functions $u_{*c\mathbf{k}}(\mathbf{r})$ in terms of the Th atomic states. Owing to the short-range nature of the HFI, relativistic effects play an important role. As a result, we use relativistic Dirac spinors for the Th atomic states. However, equation (7) is given in terms of nonrelativistic (or scalar relativistic[62]) two-component spinors. Here, for simplicity, we replace equation (7) with

$$W_{c\sigma_c v\sigma_v}^{ge}(\mathbf{k}) \equiv \frac{(2\pi)^3}{\Omega}\langle g|\mathcal{M}|e\rangle \cdot \int_\Omega d^3r u_{c\mathbf{k}\sigma_c}^\dagger(\mathbf{r})\mathcal{T}(\mathbf{r})u_{v\mathbf{k}\sigma_v}(\mathbf{r}), \tag{14}$$

in which $u_{n\mathbf{k}\sigma_n}(\mathbf{r})$ is now a four-component Dirac spinor with momentum $\mathbf{k}$ and spin projection $\sigma_n$.

The crystal function $u_{c\mathbf{k}\sigma_c}(\mathbf{r})$ (and similarly $u_{v\mathbf{k}\sigma_v}(\mathbf{r})$) may be expanded in terms of the Th atomic states of definite principle and angular momentum quantum numbers $\varphi_{njlm}(\mathbf{r})$

$$u_{c\mathbf{k}\sigma_n}(\mathbf{r}) = \sqrt{\frac{\Omega}{(2\pi)^3}} \sum_{njlm} a_{njlm}(c\mathbf{k}\sigma_n)\varphi_{njlm}(\mathbf{r}) + ..., \tag{15}$$

in which the ellipsis denotes contributions from O atomic states, which, owing to their small overlap with the Th nucleus, do not affect the HFI matrix element. Using the expansion (14), we may write

$$W_{c\sigma_c v\sigma_v}^{ge}(\mathbf{k}) = \sum_{njlm}\sum_{n'j'l'm'}a_{njlm}^*(c\mathbf{k}\sigma_c)a_{n'j'l'm'}(v\mathbf{k}\sigma_v)\widetilde{W}_{njlmnj'l'm'}^{ge}, \tag{16}$$

in which $\widetilde{W}_{njlmnj'l'm'}^{ge} = \langle g|\mathcal{M}|e\rangle \cdot \int_\Omega d^3r\varphi_{njlm}^\dagger(\mathbf{r})\mathcal{T}(\mathbf{r})\varphi_{n'j'l'm'}(\mathbf{r})$.

Given the crystal functions and the atomic wavefunctions, the expansion coefficients $a_{njlm}(\mathbf{k}\sigma_n)$ may be computed and the total HFI matrix element obtained. Here we make a simplifying assumption that only a few terms contribute substantially to the sum in equation (15). Furthermore, we shall neglect the contribution from cross terms in $|W_{c\sigma_c v\sigma_v}^{ge}(\mathbf{k})|^2$. With these simplifications, the IC rate may be written as

$$\Gamma_{IC} \approx \frac{\pi}{\hbar} \frac{1}{2I_e+1} \sum_{M_g M_e} \sum_{njlm}\sum_{n'j'l'm'} |\widetilde{W}_{njlmn'j'l'm'}^{ge}|^2$$
$$\times \sum_{\sigma_c \sigma_v} \frac{2\Omega}{(2\pi)^3}\int_{BZ} d^3k |a_{njlm}(c\mathbf{k}\sigma_c)|^2|a_{n'j'l'm'}(v\mathbf{k}\sigma_v)|^2$$
$$\delta(\varepsilon_{c\mathbf{k}} - \varepsilon_{v\mathbf{k}} - \hbar\omega_{nuc}). \tag{17}$$

Let us now consider the matrix element $W^{\mathrm{ge}}_{c\sigma_c v\sigma_v}(\mathbf{k})$ in more detail. Using the Wigner–Eckart theorem, we may write

$$\widetilde{W}^{\mathrm{ge}}_{njlmnj'\,l'm'} = \sum_{\nu=-1}^{1} (-1)^{\nu+l_g-M_g+j-m} \begin{pmatrix} I_g & 1 & I_e \\ -M_g & \nu & M_e \end{pmatrix}$$
$$\begin{pmatrix} j & 1 & j' \\ -m & -\nu & m' \end{pmatrix} \langle g \| \mathcal{M} \| e \rangle \langle njl \| \mathcal{T} \| n'j'\,l' \rangle. \tag{18}$$

Here $\langle g\|\mathcal{M}\|e\rangle \approx 0.84\mu_N$ ($\mu_N$ is the nuclear magneton) is the reduced matrix element of the nuclear magnetic moment operator (see, for example, ref. 12) and $\langle njl\|\mathcal{T}\|n'j'\,l'\rangle$ is the reduced matrix element of the rank-1 electronic HFI tensor. Th off-diagonal HFI matrix elements are at least two orders of magnitude smaller than the diagonal ones, so the most important contributions to the IC rate come from terms with $(njl) = (n'j'l')$. The reduced matrix element $\langle njl\|\mathcal{T}\|njl\rangle$ may be related to the ground state HFI constant $A_{njl}$ through[63]

$$A_{njl} = \frac{\mu_g}{I_g j} \frac{(2j)!}{\sqrt{(2j-1)!\,(2j+2)!}} \langle njl \| \mathcal{T} \| njl \rangle, \tag{19}$$

in which $\mu_g = 0.360(7)\mu_N$ is the magnetic moment of the ground nuclear state[64]. With this, the summations over $M_g$ and $M_e$ in equation (16) may be carried out, giving

$$\sum_{M_g M_e} |\widetilde{W}^{\mathrm{ge}}_{njlmnjlm'}|^2 = \frac{1}{3} \left[ \sum_{\nu=-1}^{1} \begin{pmatrix} j & 1 & j \\ -m & \nu & m' \end{pmatrix}^2 \right] \langle g\|\mathcal{M}\|e\rangle^2 \langle njl\|\mathcal{T}\|njl\rangle^2. \tag{20}$$

To proceed, we make a further assumption that the quantity in the second line of equation (16) does not depend strongly on the magnetic quantum numbers $m$ and $m'$, that is,

$$\sum_{\sigma_c \sigma_v} \frac{2\Omega}{(2\pi)^3} \int_{\mathrm{BZ}} d^3k\, |a_{njlm}(c\mathbf{k}\sigma_c)|^2 |a_{njlm'}(v\mathbf{k}\sigma_v)|^2$$
$$\delta(\varepsilon_{c\mathbf{k}} - \varepsilon_{v\mathbf{k}} - \hbar\omega_{\mathrm{nuc}})$$
$$\approx \sum_{\sigma_c \sigma_v} \frac{2\Omega}{(2\pi)^3} \int_{\mathrm{BZ}} d^3k\, |a_{njl}(c\mathbf{k}\sigma_c)|^2 |a_{njl}(v\mathbf{k}\sigma_v)|^2$$
$$\delta(\varepsilon_{c\mathbf{k}} - \varepsilon_{v\mathbf{k}} - \hbar\omega_{\mathrm{nuc}}), \tag{21}$$

then the summation over $m$ and $m'$ may be carried out analytically, finally giving

$$\Gamma_{\mathrm{IC}} \approx \frac{\pi}{3\hbar} \frac{1}{2I_e+1} \langle g\|\mathcal{M}\|e\rangle^2 \sum_{njl} \langle njl\|\mathcal{T}\|njl\rangle^2$$
$$\times \sum_{\sigma_c \sigma_v} \left[ \frac{2\Omega}{(2\pi)^3} \int_{\mathrm{BZ}} d^3k\, |a_{njl}(c\mathbf{k}\sigma_c)|^2 |a_{njl}(v\mathbf{k}\sigma_v)|^2 \right.$$
$$\left. \delta(\varepsilon_{c\mathbf{k}} - \varepsilon_{v\mathbf{k}} - \hbar\omega_{\mathrm{nuc}}) \right]. \tag{22}$$

We recognize the quantity in the square bracket as the (twice) PJDOS, which counts the number of allowed transitions at energy $\hbar\omega_{\mathrm{nuc}}$ between the atomic component $|njl\rangle$ of $|v\mathbf{k}\sigma_v\rangle$ and the atomic component $|njl\rangle$ of $|c\mathbf{k}\sigma_c\rangle$. In Fig. 3c, we present the most relevant PJDOS. These PJDOS were computed using VASP (see the next section) and use the scalar relativistic approximation of Koelling and Harmon[62]. To obtain the relativistic PJDOS, we use the procedure developed in ref. 12, that is, for any $l>0$, $|a_{n(l-1/2)l}(c\mathbf{k}\sigma_c)|^2 = |a_{n(l+1/2)l}(c\mathbf{k}\sigma_c)|^2 = |a_{nl}|^2/2$, in which $|a_{nl}|^2$ is the VASP projection. As a result, for $l>0$, the relativistic $(njl) \to (njl)$ PJDOS is a quarter of the scalar relativistic value. On the other hand, because the VASP calculation was independent of spin, the double summation over $\sigma_c$ and $\sigma_v$ in equation (21) results in a factor of $2 \times 2 = 4$. In Extended Data Table 2, we list the HFI constants $A$, the corresponding 'diagonal' values of the PJDOS at the nuclear energy for various Th

atomic states and the contributions to the IC rate. Using these values, we arrive at an estimate for the IC rate of

$$\Gamma_{\mathrm{IC}} \approx 1.2 \times 10^4 \,\mathrm{s}^{-1}, \tag{23}$$

corresponding to an IC lifetime of 80 μs.

In obtaining the estimate in equation (22), we have neglected the contribution from the cross terms in the expansion of $|W^{\mathrm{ge}}_{c\sigma_c v\sigma_v}(\mathbf{k})|^2$. Neglecting off-diagonal HFI matrix elements, the contribution from these terms reads

$$\Gamma_{\mathrm{IC}}^{\mathrm{cross\,terms}} \approx \frac{\pi}{3\hbar} \frac{1}{2I_e+1} \langle g\|\mathcal{M}\|e\rangle^2$$
$$\times \sum_{njl \neq n'j'\,l'} \langle njl\|\mathcal{T}\|njl\rangle \langle n'j'\,l'\|\mathcal{T}\|n'j'\,l'\rangle$$
$$\times \sum_{\sigma_c \sigma_v} \left[ \frac{2\Omega}{(2\pi)^3} \int_{\mathrm{BZ}} d^3k\, a^*_{njl}(c\mathbf{k}\sigma_c) a_{njl}(v\mathbf{k}\sigma_v) a_{n'j'\,l'}(c\mathbf{k}\sigma_c) \right.$$
$$\left. a^*_{n'j'\,l'}(v\mathbf{k}\sigma_v) \delta(\varepsilon_{c\mathbf{k}} - \varepsilon_{v\mathbf{k}} - \hbar\omega_{\mathrm{nuc}}) \right]. \tag{24}$$

Heuristically, we expect the different terms in equation (23) to have different phases and, thus, their total sum to be small. A more detailed calculation in which the expansion coefficients $a_{njl}(c\mathbf{k}\sigma_c)$ and $a_{njl}(v\mathbf{k}\sigma_v)$ are computed explicitly is needed to precisely gauge the accuracy of this approximation. This is the subject of a future work. Here we assume that the expansion coefficients have the same phase (and may thus be taken as all real) and estimate the maximum contributions from the cross terms. In this case, the quantity in the square bracket of equation (23) is a 'cross-PJDOS', which is obtained by summing over the delta functions weighted by the product of square roots of $|a_{njl}|^2$. The maximum values for the cross terms are listed in Extended Data Table 3. The sum of these cross contributions amounts to

$$\Gamma_{\mathrm{IC}}^{\mathrm{max\,cross\,terms}} \approx 2.6 \times 10^4 \,\mathrm{s}^{-1}. \tag{25}$$

The estimated maximum IC rate is therefore

$$\Gamma_{\mathrm{IC}}^{\mathrm{max}} \approx 3.8 \times 10^4 \,\mathrm{s}^{-1}, \tag{26}$$

corresponding to an IC lifetime of 30 μs.

A further reduction in the IC lifetime is very plausible if we that the relative error in making the non-relativistic approximation $|a_{n(l-1/2)l}(c\mathbf{k}\sigma_c)|^2 = |a_{n(l+1/2)l}(c\mathbf{k}\sigma_c)|^2 = |a_{nl}|^2/2$ is approximately $(\alpha Z)^2$, that is, about 40% for $Z = 90$ of Th. Because relativistic contraction generally increases the electron density near the origin, the IC rate including relativistic effects may be larger than its non-relativistic counterpart by a factor of about $1.4^2 = 2$, thus further reducing the estimated IC lifetime to 15 μs.

## Computational methods for electronic structure theory

Calculations were performed with VASP version 6.4.2 (ref. 65) using the PAW[66] method. The structure of $ThO_2$ was optimized with DFT in the conventional unit cell using the PBE[67] functional, 6-6-6 $\Gamma$-centred $k$-mesh and a 500-eV plane wave cut-off. Our computed lattice parameter is 5.617 Å, which matches very well with experimental measurements of 5.597 Å (ref. 68). Subsequent electronic structure calculations used the optimized structure in the primitive cell representation.

Parameters for $G_0W_0$ calculations[69–71], specifically the $k$-mesh, plane wave cut-off energy and the number of frequency grid points (NOMEGA tag in VASP), were tested for convergence of the bandgap. The $k$-mesh was tested with a 400-eV cut-off and NOMEGA = 80, the cut-off was tested with a 6-6-6 $k$-mesh and NOMEGA = 80 and NOMEGA was tested with a 400-eV cut-off and a 4-4-4 $k$-mesh. The number of unoccupied bands was 812 (there are 12 occupied bands). The results of these tests are shown in Extended Data Tables 4–6. On the basis of these results,

further $G_0W_0$ calculations were done with an 8-8-8 $k$-mesh, a 500-eV plane wave cut-off and 80 frequency grid points.

Our converged bandgap of 6.20 eV agrees well with single-crystal and thin-film measurements of 5.9 eV (refs. 42,43), although we note that a range of bandgaps has been reported on the basis of a range of experimental samples (thin films, nanoparticles, single crystals)[72–76] because the measured absorption spectrum can be strongly influenced by morphology, defects and the effects of irradiation.

The BSE is a two-particle Green's function formalism to explicitly account for electron–hole interactions in electronic excited states[77,78]. The $G_0W_0$ + BSE parameters were converged with respect to the predicted absorption spectrum. The tested parameters were the highest excitation energy (OMEGAMAX) and the number of occupied and unoccupied bands (NBANDSO and NBANDSV, respectively) considered in the BSE calculation. Absorption spectra computed with various settings for these methods are shown in Extended Data Fig. 2a,b. On the basis of these tests, further $G_0W_0$ + BSE calculations were carried out with OMEGAMAX = 20 eV and (NBANDSO, NBANDSV) = (8, 16).

We can validate our method against published experimental data by computing the dielectric function with $G_0W_0$ + BSE. Our computed data, shown in Extended Data Fig. 2c,d, are a good match with the spectroscopic data in ref. 72. Some differences in the imaginary part of the dielectric function $\varepsilon_2$, namely the rate at which $\varepsilon_2$ increases with increasing energy from 5 to 8 eV and the low-energy non-zero 'tail' in the experimental spectrum, we attribute to the presence of defects in the real crystal, as noted by the authors of the experimental study.

We also compute the absorption spectrum of $ThO_2$ with $G_0W_0$ + BSE, as shown in Extended Data Fig. 2e. The value at the nuclear transition energy is $1 \times 10^6$ cm$^{-1} \equiv 0.1$ nm$^{-1}$, in agreement with ellipsometric measurements[35].

Absorption spectra and dielectric functions were processed from VASP output with VASPKIT[79].

## Isomer shift

The isomer shift originates from differences in the nuclear charge distribution between the ground and excited nuclear states. The value of the shift depends on the local electronic environment of $^{229}$Th. Here we follow the formalism[40] that combines relativistic many-body atomic-structure methods with periodic DFT to evaluate the isomer shifts in $^{229}ThO_2$. The scalar relativistic periodic DFT reproduces valence band properties, whereas relativistic atomic-structure methods capture the essential core-electron relaxation effects. The valence band contribution to the isomer shift in $^{229}$Th solid-state hosts is expressed as[40]

$$\delta E_{\text{iso}}^{\text{VB}} = \sum_{\ell} \text{IPDOS}_{\ell}\, \delta \varepsilon_{\ell}^{\text{iso}}(\text{Th}^{3+}), \qquad (27)$$

in which $\text{IPDOS}_{\ell}$ denotes the integrated projected (on $^{229}$Th) valence band density of states for angular momentum $\ell$ and $\delta\varepsilon_{\ell}^{\text{iso}}(\text{Th}^{3+})$ is the isomer shift for the lowest-energy valence orbitals of $\text{Th}^{3+}$ of angular momentum $\ell$. The valence band isomer shift is to be added to the isomer shift in $^{229}\text{Th}^{4+}$ ions; this contribution remains constant across a wide range of materials. Extended Data Table 7 presents the calculated isomer shifts for $^{229}ThO_2$ for three different methods. Compared with the PBE and MBJ[80,81] methods, the $G_0W_0$ method includes self-energy correction and, as discussed earlier, we consider it of a higher quality.

Extended Data Table 8 presents the calculated isomer shifts $\delta E_{\text{iso}}^{\text{VB}}$, the corresponding nuclear clock frequencies $\nu$ of $^{229}$Th and their offsets $\Delta \nu$ relative to the $ThO_2$ reference for a range of solid-state hosts.

## Determination of clock stability

We estimate the clock stability by[82]

$$\sigma = \frac{1}{2\pi QS}\sqrt{\frac{T_e + T_c}{\tau}}, \qquad (28)$$

in which $Q = f_0/\Delta f$ is the transition quality factor, $S$ is the signal-to-noise ratio, $T_e$ is the excitation time, $T_c$ is the electron collection time and $\tau$ is the averaging time. As stated in the text, we assume that $\Delta f$ is dominated by the homogeneous lifetime broadening owing to IC and use $\Delta f = 16$ kHz. For our analysis, we assume $T_e = T_c = 12$ μs and a shot-noise-limited signal-to-noise ratio given by $S = \sqrt{N_{\text{det}}}$, in which $N_{\text{det}}$ is given by

$$N_{\text{det}} = N_e \times (1 - e^{-T_c/\tau_{\text{IC}}}), \qquad (29)$$

$N_e$ ($\approx 2.5 \times 10^7$) is the total number of $^{229}$Th nuclei excited, $\eta = 0.5$ is the electron detection efficiency in the clock system and $\tau_{\text{IC}} = 12$ μs. $N_e$ is calculated assuming that the probe laser linewidth is much smaller than the lifetime-limited transition linewidth, and the probe laser power is 100 μW. From this, we obtain a projected clock performance of about $2 \times 10^{-18}/\sqrt{\tau}$.

## Data availability
The data that support the findings of this study are available from the corresponding author on request.

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

**Acknowledgements** This work was supported by NSF awards PHYS-2013011, PHY-2412869, PHY-2513134, and PHY-2207546 and ARO award W911NF-11-1-0369. P.G.T. and K.S.

acknowledge support by BaCaTeC (grant 7 [2029-2]) and funding from the European Research Council (ERC) under the European Union's Horizon 2020 Research and Innovation Program ('ThoriumNuclearClock', Grant Agreement No. 856415). E.R.H. acknowledges institutional support by the NSF QLCI award OMA-2016245. This work used Bridges-2 at Pittsburgh Supercomputing Center through allocation PHY230110 from the Advanced Cyberinfrastructure Coordination Ecosystem: Services & Support (ACCESS) programme, which is supported by National Science Foundation grants #2138259, #2138286, #2138307, #2137603 and #2138296. H.B.T.T. and D.A.R. acknowledge support from the Laboratory Directed Research and Development programme under projects 20258345CT-IMS and 20260021DR, as well as computational resources provided by the Institutional Computing Program through the Center for Integrated Nanotechnologies, a DOE BES user facility, at Los Alamos National Laboratory.

**Author contributions** R.E., J.E.S.T., C.S. and E.R.H. constructed the experimental chamber and performed the spectroscopy measurement. H.W.T.M., H.B.T.T., U.C.P., D.A.R. and A.D. performed theory calculations. M.C.A. produced and characterized the sample. L.v.d.W., B.S., K.S. and P.G.T. calibrated and provided detectors. All authors contributed to the manuscript.

**Competing interests** E.R.H. declares a provisional patent (U.S. provisional patent application no. 63/814,873) related to the findings presented in this work.

**Additional information**
**Correspondence and requests for materials** should be addressed to Eric R. Hudson.

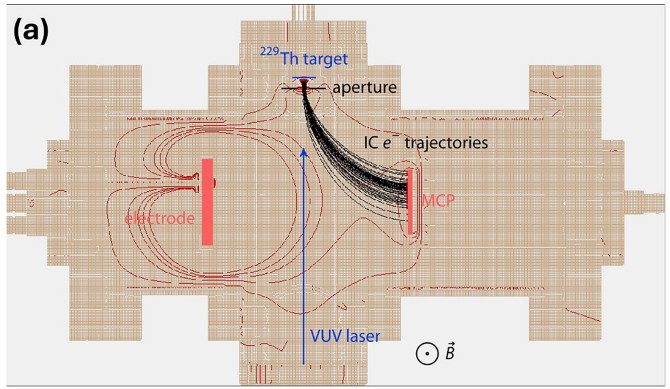

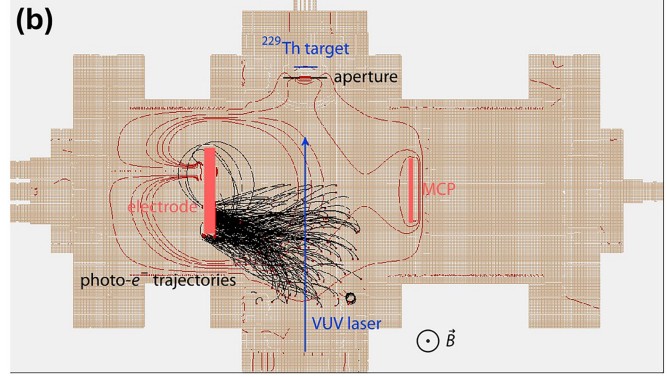

**Extended Data Fig. 1 | Electron trajectory simulations.** SIMION simulations of electron trajectories under the voltage biasing and static magnetic field used in the experiment. **a**, Trajectories of IC electrons from the target region. The magnetic field bends them towards the detection MCP through a saddle in the electric potential. **b**, The trajectories of photoelectrons generated by VUV scattered light at random locations in the chamber. Most either crash into the chamber walls or end up falling into the electrode biased at +2,500 V.

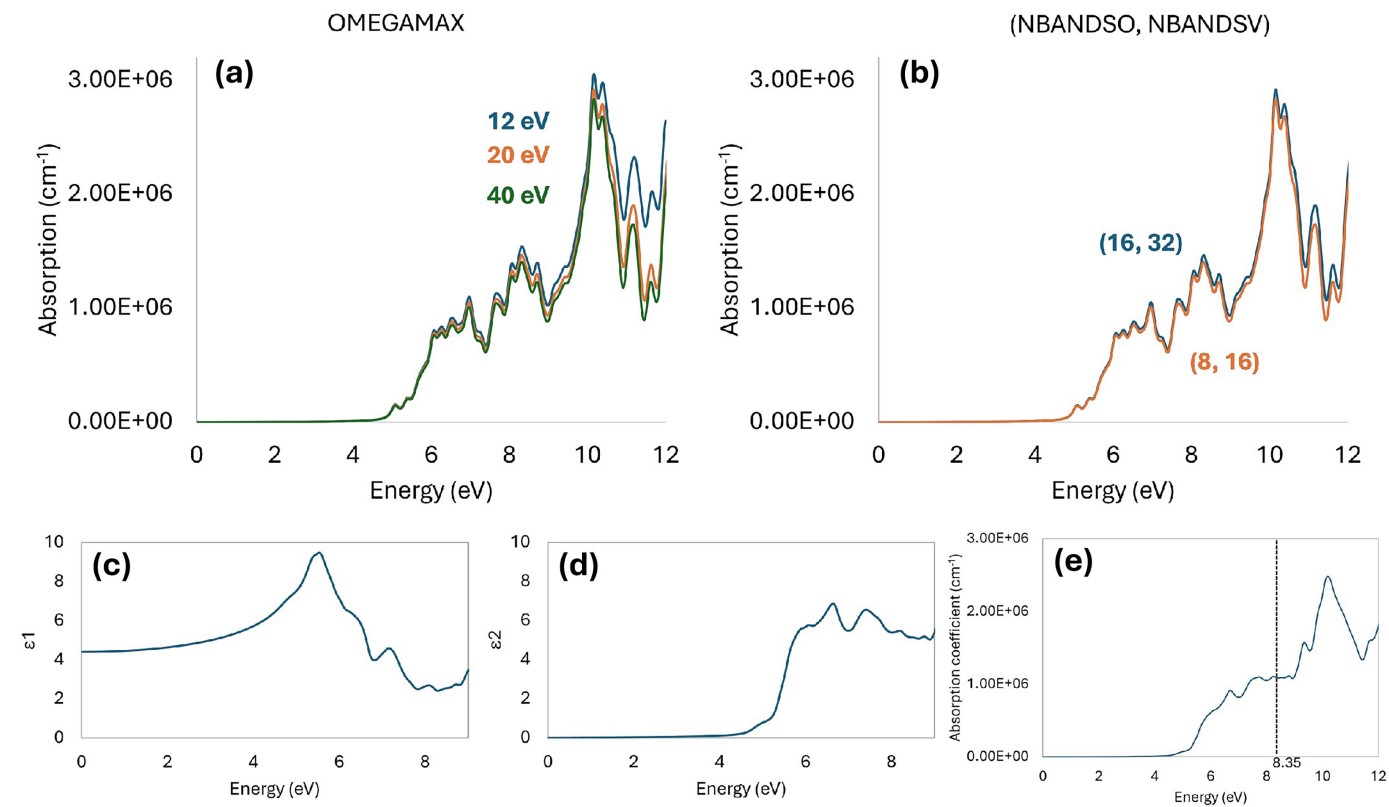

**Extended Data Fig. 2 | ThO₂ optical properties through DFT. a,b**, Absorption spectra of ThO₂ computed with $G_0W_0$ + BSE using different OMEGAMAX (**a**) and (NBANDSO, NBANDSV) (**b**) settings. The meanings of these parameters are given in the text. **c,d**, Real ($\varepsilon_1$) and imaginary ($\varepsilon_2$) parts of the frequency-dependent dielectric function for bulk ThO₂ computed with $G_0W_0$ + BSE. **e**, Absorption spectrum of bulk ThO₂ computed with $G_0W_0$ + BSE.

**Extended Data Table 1 | Parameter values and estimated relative errors used in the calculation of expected IC signal**

| Parameter | Value | Rel. Err. |
|---|---|---|
| $n_{\mathrm{Th}}$ | $1.71 \times 10^{22}$ cm$^{-3}$ | 0 (Ref. [48]) |
| $L$ | 10 nm | 0.7* |
| $\Gamma_L$ | $2\pi \times 12$ GHz | 0.25 |
| $\alpha$ | 0.1 nm$^{-1}$ | 0 (Ref. [35]) |
| $n$ | 2.34 | 0 (Ref. [35]) |
| $t_i$ | 6 $\mu$s | 1e-4 |
| $\tau_{\mathrm{rad}}$ | 1800 s | 0.07 |
| $\tau_{\mathrm{IC}}$ | 10 $\mu$s | 0.1 |
| $T\eta_{\mathrm{e}}$ | $1 \times 10^{-4}$ | 0.7* |
| $\eta_{\mathrm{c}}$ | 0.7 | 0.35* |

Relative uncertainties labelled by an asterisk follow a log-normal distribution and should be read as $\ln(X/\overline{X})$. 0 error entries correspond to assumed values from the literature.

**Extended Data Table 2 | Magnetic dipole $A$ HFI constants for low-lying levels of $^{229}\text{Th}^{3+}$, the corresponding values of PJDOS at the nuclear energy about 8.35 eV and the contributions to the IC rate**

| | $A$ (MHz) | PJDOS($\hbar\omega_{\text{nuc}}$) (1/eV) | Contribution to $\Gamma_{\text{IC}}$ ($s^{-1}$) |
|---|---|---|---|
| $7s_{1/2} - 7s_{1/2}$ | $+6249$ | $5.0 \times 10^{-5}$ | 431 |
| $6p_{1/2} - 6p_{1/2}$ | $+8604$ | $4.3 \times 10^{-4}$ | 6955 |
| $6p_{3/2} - 6p_{3/2}$ | $+426$ | $4.3 \times 10^{-4}$ | 170 |
| $6d_{3/2} - 6d_{3/2}$ | $+155.3$ | $5.8 \times 10^{-3}$ | 307 |
| $6d_{5/2} - 6d_{5/2}$ | $-12.6$ | $5.8 \times 10^{-3}$ | 7.1 |
| $5f_{5/2} - 5f_{5/2}$ | $+82.0$ | $5.3 \times 10^{-2}$ | 2731 |
| $5f_{7/2} - 5f_{7/2}$ | $+31.4$ | $5.3 \times 10^{-2}$ | 961 |

The $A$ constants for 7s and 6p states are computed using our relativistic atomic-structure code with random phase approximation and perturbative Brueckner orbital corrections. The $A$ constants for the 6d and 5f states are taken from experiments[83].

**Extended Data Table 3 | Maximum contributions to the IC rate from cross terms arising from the expansion in equation (15), assuming that all of the expansion coefficients are in phase**

| $n'j'l' - njl$ | "cross-PJDOS" $(\hbar\omega_{\mathrm{nuc}})$ $(1/\mathrm{eV})$ | Maximum contribution to $\Gamma_{\mathrm{IC}}$ $(s^{-1})$ |
|---|---|---|
| $7s_{1/2} - 6p_{1/2}$ | $5.0 \times 10^{-5}$ | 1177 |
| $7s_{1/2} - 6p_{3/2}$ | $5.0 \times 10^{-5}$ | 184 |
| $7s_{1/2} - 6d_{3/2}$ | $2.7 \times 10^{-4}$ | 366 |
| $7s_{1/2} - 6d_{5/2}$ | $2.7 \times 10^{-4}$ | 56 |
| $7s_{1/2} - 5f_{5/2}$ | $5.0 \times 10^{-4}$ | 670 |
| $7s_{1/2} - 5f_{7/2}$ | $5.0 \times 10^{-4}$ | 398 |
| $6p_{1/2} - 6p_{3/2}$ | $4.3 \times 10^{-4}$ | 2178 |
| $6p_{1/2} - 6d_{3/2}$ | $8.8 \times 10^{-4}$ | 1635 |
| $6p_{1/2} - 6d_{5/2}$ | $8.8 \times 10^{-4}$ | 248 |
| $6p_{3/2} - 6d_{3/2}$ | $8.8 \times 10^{-4}$ | 256 |
| $6p_{3/2} - 6d_{5/2}$ | $8.8 \times 10^{-4}$ | 39 |
| $6p_{1/2} - 5f_{5/2}$ | $4.0 \times 10^{-3}$ | 7381 |
| $6p_{1/2} - 5f_{7/2}$ | $4.0 \times 10^{-3}$ | 4379 |
| $6p_{3/2} - 5f_{5/2}$ | $4.0 \times 10^{-3}$ | 1156 |
| $6p_{3/2} - 5f_{7/2}$ | $4.0 \times 10^{-3}$ | 686 |
| $6d_{3/2} - 6d_{5/2}$ | $1.1 \times 10^{-2}$ | 93 |
| $6d_{3/2} - 5f_{5/2}$ | $5.8 \times 10^{-3}$ | 1185 |
| $6d_{3/2} - 5f_{7/2}$ | $5.8 \times 10^{-3}$ | 703 |
| $6d_{5/2} - 5f_{5/2}$ | $5.8 \times 10^{-3}$ | 180 |
| $6d_{5/2} - 5f_{7/2}$ | $5.8 \times 10^{-3}$ | 107 |
| $5f_{5/2} - 5f_{7/2}$ | $5.3 \times 10^{-2}$ | 3204 |

The $6d_{5/2}$–$6d_{5/2}$ HFI matrix elements carry a sign opposite to all the others. This sign difference is also ignored; in any case, the contributions from cross terms involving $6d_{5/2}$ orbitals are relatively small.

**Extended Data Table 4 | Convergence of the $G_0W_0$ bandgap with respect to $k$-mesh**

| mesh | spacing (Å$^{-1}$) | band gap (eV) |
|------|--------------------|---------------|
| 6-6-6 | 0.05 | 6.31 |
| 7-7-7 | 0.045 | 6.21 |
| 8-8-8 | 0.04 | 6.18 |
| 9-9-9 | 0.035 | 6.22 |

**Extended Data Table 5 | Convergence of the $G_0W_0$ bandgap with respect to plane wave cut-off**

| Cut-off (eV) | band gap (eV) |
|---|---|
| 400 | 6.31 |
| 500 | 6.30 |
| 600 | 6.29 |

**Extended Data Table 6 | Convergence of the $G_0W_0$ bandgap with respect to frequency grid points (NOMEGA)**

| NOMEGA | band gap (eV) |
|---|---|
| 8 | 7.45 |
| 20 | 6.36 |
| 40 | 6.21 |
| 80 | 6.18 |

**Extended Data Table 7 | Integrated partial densities of states, IPDOS$_\ell$, and the products, $\delta E_{\mathrm{iso},\ell}^{\mathrm{VB}}$ = IPDOS$_\ell \delta \varepsilon_\ell^{\mathrm{iso}}$(Th$^{3+}$), obtained with the three electronic structure methods**

| $\ell$ | $G_0 W_0$ | | MBJ | |
| --- | --- | --- | --- | --- |
| | IPDOS$_\ell$ | $\delta E_{\mathrm{iso},\ell}^{\mathrm{VB}}$ (a.u.) | IPDOS$_\ell$ | $\delta E_{\mathrm{iso},\ell}^{\mathrm{VB}}$ (a.u.) |
| $s$ | 0.119 | $+1.62 \times 10^{-8}$ | 0.078 | $+1.07 \times 10^{-8}$ |
| $p$ | 0.435 | $-1.69 \times 10^{-9}$ | 0.408 | $-1.58 \times 10^{-9}$ |
| $d$ | 0.759 | $-2.42 \times 10^{-8}$ | 0.688 | $-2.19 \times 10^{-8}$ |
| $f$ | 0.534 | $-4.25 \times 10^{-8}$ | 0.432 | $-3.43 \times 10^{-8}$ |
| $\delta E_{\mathrm{iso}}^{\mathrm{VB}}$ (MHz) | $-343$ | | $-310$ | |

The isomer shift of the solid-state Th clock, $\delta E_{\mathrm{iso}}^{\mathrm{VB}}$ (bottom row), is the summation over the orbital $\ell$ defined in equation (26).

**Extended Data Table 8 | Nuclear clock frequencies $\nu$ obtained from the isomer shifts $\delta E_{\mathrm{iso}}^{\mathrm{VB}}$ and their offsets $\Delta\nu$ relative to $ThO_2$ computed with MBJ**

| Host | $\delta E_{\mathrm{iso}}^{\mathrm{VB}}$ (MHz) | $\nu$ (MHz) | $\Delta\nu$ (MHz) |
|---|---|---|---|
| $ThO_2$ | $-310$ | $2\,020\,407\,338(70)$ | $0$ |
| $CaF_2$ F–90°–F, Ref. [41] | $-264$ | $2\,020\,407\,384(70)$ | $+46$ |
| $LiSrAlF_6$, Ref. [41] | $-234$ | $2\,020\,407\,414(70)$ | $+76$ |

The absolute frequencies are referenced to the free-ion value $\nu(Th^{4+})$=2,020,407,648(70) MHz taken from ref. 40.