## [Peer Review File · Nature]

Laser-Based Conversion Electron Mössbauer Spectroscopy of $^{229}\text{ThO}_2$

Corresponding Author: Professor Eric Hudson

Version 0:

Reviewer comments:

Referee #1

(Remarks to the Author)

Recent achievement of a major milestone in the topical research field aimed at studies of the uniquely low energy state (8.4 eV) nuclear isomer, namely, its laser excitation, led to an explosive development of the field. This fast progress indicates that highly anticipated applications of this isomer, such as building the Mössbauer nuclear clock, development of super sensors for material studies and search for dark matter and time-dependence of the fundamental constants, are now right on the horizon.

This paper opens a new avenue in experimental investigation of the ^{229}Th solid compounds, significantly broadening the scope of the current research in this field. While, so far, such studies have been focused on the ^{229}Th doped metal fluoride crystals with the band gap exceeding the isomer energy (8.4 eV), this work for the first time experimentally investigates laser excitation of ^{229}Th in a narrow band host (~ 6 eV), namely, in $^{229}\text{ThO}_2$.

While an internal conversion channel was eliminated or significantly suppressed in the wide band gap materials, it greatly dominates the radiative channel in the narrow band gap materials, reducing the isomer lifetime by 8 orders of magnitude compared to the radiative decay time. Measuring the number of electrons ejected from the ThO_2 surface because of the internal conversion process as the function of frequency detuning of laser radiation from the resonant nuclear transition allows for monitoring the spectrum of the nuclear absorber dependent on the parameters of the host environment. The proof of principle of such novel laser Conversion Electron Mössbauer Spectroscopy (CEMS) has been demonstrated in this work for the first time. The possibility to increase the stability of ^{229}Th nuclear clocks using such narrow band gap solids (compared to the clocks based on the wide gap ^{229}Th doped metal fluoride crystals possessing large $\sim 10\text{kHz}$ inhomogeneous broadening due to dipole-dipole interaction between ^{229}Th and F nuclear spins) by reducing an interrogation time due to accelerated nuclear decay has been also outlined. All these three mutually related factors (new class of materials, CEMS and a way to achieve high stability) point together to the breakthrough nature of this research, qualifying it for publication in Nature.

At the same time, several comments and questions need to be addressed in the paper.

1. CEMS has been widely used by Mössbauer community for over 50 years as a sensitive tool for studying the nuclear environment (see for example, M. J. Tricker, J.M. Thomas, A.P. Winterbottom, Conversion-electron Mössbauer spectroscopy for the study of solid surfaces, *Surface Science*, 45, 601, 1974). We believe this fact needs to be acknowledged together with the novelty introduced by this work into this technique, namely, using the laser source instead of the radioactive sources used so far.

2. In connection with the above comment an important question arises. The ability to record the spectral line of the nuclear absorber by means of CEMS relies on using spectrally narrow source (compared to the spectral width of the absorber). However, the linewidth of a laser used in this work was a million times broader than the linewidth of the nuclear transition. It seems not much spectral information can be recovered from this type of measurement except the resonance position. The full potential of this technique can be realized in the future when sufficiently narrow sources (such as based on intracavity HHG) become available.

3. It is emphasized several times that CEMS, unlike fluorescence spectroscopy, allows to study materials whose work function is less than the nuclear transition energy. It may be true for the specific case of ^{229}Th compounds, where an internal conversion factor is enormously high ($\sim 10^8$ - 10^9) so that the fluorescence is too weak to be registered. But in

general, both fluorescence and CEMS are widely used to study materials whose work function is less than the nuclear transition energy. A common example is ^{57}Fe where an internal conversion factor is only 9. The other example is ^{45}Sc , where an internal conversion factor is ~ 400 . Perhaps, this point needs to be clarified to avoid misunderstanding.

4. It would be beneficial for the readers to mention the recent work on resonant excitation of the long-lived ^{45}Sc nuclear isomer (Yu. Shvyd'ko, et al., Resonant X-ray excitation of the nuclear clock isomer ^{45}Sc , Nature 622, 471 (2023)). Indeed, in addition to ^{229}Th it is the only one more potential nuclear clock isomer which was directly resonantly excited with x-ray laser. Besides, similar to this work, the technique used for detection of the resonant excitation of ^{45}Sc was based on the internal conversion process, though instead of direct counting the conversion electrons, the characteristic $K\alpha$ and $K\beta$ x-ray photons at ~ 4 keV caused by internal conversion process were recorded.

5. The statement about improvement of nuclear clock stability by reducing interrogation time due to reduction of T1 by an internal conversion process in the narrow band gap materials, pointed out several times in the text, may cause confusion and need clarification. Indeed, based on this statement one may conclude that using such materials is the most advantages for producing stable nuclear clock, which is in an obvious contradiction with a common sense to use quantum transitions from the long-lived states (and hence narrowest possible linewidth) to maximize clock's accuracy and stability. Apparently, using narrow band gap materials for nuclear clock (resulting in the life-time line broadening to 16 kHz) is more profitable than using a wide band gap materials with large inhomogeneous line broadening (such as metal fluorides with ~ 10 kHz linewidth). But it is not beneficial at all as compared to using the wide band gap spinless materials with very narrow spectral lines (which were discussed in the work of the same group [4]) because Q factor in these materials would be dramatically higher than in the narrow band gap materials resulting in higher clock stability.

6. We believe the statement "Further, because ThO_2 can be made from spinless isotopes [4] and the internal conversion decay process reduces the isomeric state lifetime to only ~ 10 μs , allowing ~ 108 relative reduction in clock interrogation time, a conversion-electron-based nuclear clock could lead to a ~ 104 improvement in clock instability" needs to be either explained or corrected. Indeed, the role of spinless host in this context is not clear. Once the linewidth is determined by the life-time broadening (~ 16 kHz), any narrow band gap host (spinless or not) would lead to the same result (i.e., stability improvement compared to the wide band gap materials with similarly large inhomogeneous broadening due to quenching of nuclear decay).

Olga Kocharovskaya and Xiwen Zhang

Referee #2

(Remarks to the Author)

I co-reviewed this manuscript with one of the reviewers who provided the listed reports.

Referee #3

(Remarks to the Author)

The manuscript by Elwell and coworkers presents a significant and timely experimental advance: the use of a VUV laser to perform ^{229}Th Conversion Electron Mössbauer Spectroscopy (CEMS) on ThO_2 . The authors correctly highlight the primary challenge of performing nuclear resonance spectroscopy in materials where the nuclear transition energy (8.3 eV) is greater than the material's band gap (in this case ~ 6 eV). The successful application of laser-driven CEMS to overcome this fundamental obstacle is a commendable achievement and represents a valuable contribution to the field.

The manuscript makes two high-impact claims: (1) that this technique opens the door to studying the ^{229}Th transition in a new class of low-bandgap materials, and (2) that this approach could lead to a substantial improvement in nuclear clock stability. The first claim is well-supported and compelling. The demonstrated ability to acquire a clean spectrum where traditional methods would fail is a clear strength of this work.

However, the second, and arguably more significant, is that the CEMS approach can lead to a higher sampling rate for these types of clocks and thus improvement in stability over other proposed Th-based clocks. However, I think immediate comparisons are premature. Bandgaps tend to correlate with Debye temperatures so materials amenable to CEMS (low bandgap) may then have Debye temperatures low enough that second order doppler shifts become problematic towards clock stability whereas conversely high band-gap materials do not face this issue as significantly. Establishing the magnitude of SODS in Thoria is outside of the scope of this manuscript but the concept should at least be addressed in the context of the atomic clock performance to prevent any misleading of the readers. The recent PRL 134,113801 (2025) from Higgins et al. working on understanding the second order doppler effect towards a Th-229 atomic clock touches directly on this topic. I think this latter publication should be cited in this manuscript and discussed in this context. Arguably it could be made to support the case here.

Moreover, I think the calculations in the manuscript need substantially more experimental validation and the evidence offered in the manuscript towards this end is nearly misleading.

Overall, I think the experimental work is heroic but specialized. The mathematical derivations are sound, but the quantum chemical calculations are poorly validated. I think there is substantial impact limited to a smaller community than the general Nature readership. Therefore, I believe these authors would better reach their target audience in another journal.

Below are specific points that I think if addressed would strengthen the manuscript.

- As noted above, the discussion of clock stability is incomplete without a thorough treatment of the Second-Order Doppler Shift. The authors should address the topic of SODS as a potential source of frequency instability. They should provide an estimate for the SODS-induced temperature sensitivity in this system and discuss the demanding technical requirements for its stabilization. This will provide a more realistic assessment of the proposed clock's potential for the readers.
- The calculations need more experimental validation. The authors mention the 6.18 eV calculated bandgap is in good agreement with literature values but the bandgap from Mock et al. cited in this manuscript is substantially lower (5.4 eV) than the calculated bandgap here. What in the calculation is the source of this difference? The authors argue their calculated

frequency dependent dielectric functions are comparable to Ref. 37 and thus experimentally validate the calculations, but inspection of Fig 4 in that reference demonstrates substantial qualitative differences that require explanation or alternative validation of these calculations. The other experimental validation offered is the absorption coefficient of 0.1 nm^{-1} calculated in agreement with ellipsometric measurements from figure 3.4 in a seemingly non-peer reviewed capstone project write-up. That figure has four measurements in it giving a distribution of values arguably around 0.1 nm^{-1} the spectra also lack any of the qualities in the calculated spectrum of Figure 6 in this manuscript in this same region. I don't diminish the reference being a capstone project but there must be a better reference to ground these calculations in or an explanation resolving why there are so many differences from the spectra calculated here. Even comparing the geometry optimized structure to the actual crystal structure would be a meaningful step in the right direction.

- Along these lines, the authors use a 500 eV plane wave cutoff for the geometry optimization but then use a 400 eV for the G0W0 calculations. Table IV might indicate a reason for this choice, but it should be more plainly justified – or the two calculations should be done with the same cutoff.
- Figure 8 features no x-axis label.
- I find the thought of doing Th-229 nuclear resonance vibrational spectroscopy at sub-meV resolution a very good idea and this group should pursue that in the future.

Referee #4

(Remarks to the Author)

The ^{229}Th nucleus is a prime candidate for developing nuclear optical clocks and probing new physics beyond the Standard Model. Current approaches for building nuclear optical clocks include: (i) The ionic approach – based on trapped ^{229}Th ions, which may achieve the highest precision but suffers from extremely weak nuclear excitation and fluorescence signals (not yet observed experimentally) due to the limited number of trapped ions (~ 1000). (ii) The doped-crystal approach – based on ^{229}Th -doped crystals (e.g., $\text{Th}:\text{CaF}_2$ and $\text{Th}:\text{LiSrAlF}_6$), which provides stronger signals owing to the large number of doped nuclei but the precision may be limited by electronic and phononic interactions in the crystal. (iii) The thin-film approach – based on $^{229}\text{ThF}_4$ thin films, which requires less ^{229}Th material but the clock performance may depend sensitively on film geometry. Both the doped-crystal and thin-film approaches require host materials with high band gaps ($>8.4 \text{ eV}$) and vacuum ultraviolet (VUV) transparency. All existing approaches also face the challenge of long clock interrogation times.

This manuscript experimentally demonstrates a new and promising approach based on laser-induced internal-conversion (IC) electrons in $^{229}\text{ThO}_2$ thin films, the band gap of which is smaller than the nuclear transition energy. In contrast to $^{229}\text{ThF}_4$ thin films, the isomeric state in $^{229}\text{ThO}_2$ decays much faster to the ground state via IC, promoting valence-band electrons to the conduction band and ultimately ejecting electrons that can be detected.

Key experimental findings include:

- (i) A measured nuclear transition frequency of 2020407 GHz, consistent with previous laser excitation experiments in $^{229}\text{Th}:\text{LiSrAlF}_6$.
- (ii) A reduced isomeric-state lifetime of $\sim 10 \mu\text{s}$, much shorter than the radiative decay lifetime, enabling a significant reduction in clock interrogation time.

The authors also present a comprehensive theoretical analysis of the IC decay rate, including electronic-structure calculations, hyperfine interactions, and isomer shifts in solids.

The results are exciting and represent a significant step forward in nuclear clock development. The concept of an IC-based solid-state nuclear clock is demonstrated to be both architecturally sound and technically feasible. This approach has great potential for the miniaturization of nuclear clocks and for enabling practical applications. Given its expected high impact and broad interdisciplinary interest, I recommend publication in Nature after the following questions and comments are addressed.

- (i) In the abstract: “this technique is compatible with materials whose work function is less than the nuclear transition energy...” – please provide the definition and value of the work function of the ThO_2 thin film. In Fig. 3(c), the authors present the calculated Th PDOS, and the energy gap between the top of the conduction band and the Fermi level is clearly larger than 8.4 eV.
- (ii) It is unclear whether the observed electron emission follows: (a) IC directly promoting valence-band electrons to the conduction band, followed by escape to vacuum, or (b) photoexcitation of electrons to the conduction band by the laser pulse, followed by IC promotion of these conduction-band electrons into vacuum. The manuscript discusses only process (a). Are there experimental data or theoretical arguments ruling out process (b)?
- (iii) The derivation of the IC rate is comprehensive; however, some of the approximations used require further explanation. In the second paragraph of Page 14, the authors state that “we shall neglect the contribution cross terms in $|W_{(c\sigma_c v\sigma_v)}^{\text{ge}}(\mathbf{k})|^2$ ”, which ultimately alters the order of summation and modulus square operations. This approximation is not obviously reasonable. Correspondingly, the calculated IC rate is about $1.3 \times 10^4 \text{ s}^{-1}$, which is lower than the observed IC decay rate of approximately 10^5 s^{-1} . Does the neglect of the cross terms lead to a significant reduction in the calculated IC rate? This assumption needs to be evaluated in the calculation. Furthermore, in the first paragraph of Page 15, the expansion coefficients are set as $|a_{(n-1/2)}|^2 = |a_{(n+1/2)}|^2 = |a_{(n)}|^2/2$. This approximation is introduced to simplify the treatment of spin-orbit coupling. How does it quantitatively affect the calculated IC decay rate? Please provide a justification of its validity.
- (iv) The authors predict a projected clock instability of about 2×10^{-18} at 1 s averaging. Please provide a more detailed evaluation in the Supplemental Information.

Version 1:

Reviewer comments:

Referee #1

(Remarks to the Author)

We appreciate the response of the authors to our comments and suggestions and confirm our strong recommendation for publication these breakthrough results in Nature.

Olga Kocharovskaya and Xiwen Zhang

Referee #3

(Remarks to the Author)

The authors here are utilizing conversion electrons Mossbauer spectroscopy (CEMS) to probe the nuclear excited state of thorium utilizing a benchtop laser. The results have implications for the development of nuclear clocks and access to the study of low bandgap Th-229 material.

These authors have addressed my previous concerns with the manuscript and the (in my opinion) the concerns of the other referees during that submission. Most of their modifications to the text have been additions and clarifications for the benefit of the reader and to help the potentially broad audience of Nature. They specifically entreated the second-order-doppler effect contributions to the method. They've made the computational section more robust and made stronger connections to experimental data to validate their calculations. I think this work is amenable to publication now.

I was previously against, but am now ambivalent about, whether Nature is the ideal journal for this work. I think they've made the work approachable to general readers in proximal fields. It is not obvious to me that the manuscript will resonate with the target audience in Nature. I do agree it is a new field that this work may open up - but a highly specialized one that may not have the global impact that Nature content tends towards.

Referee #4

(Remarks to the Author)

In the revised manuscript, the authors have satisfactorily addressed the questions I raised in my previous report.

As requested by the Editor, I also provide my view on the authors' responses to the comments of Reviewer #3. Reviewer #3 rightly acknowledged that this work opens the door to studying the ^{229}Th transition in a new class of low-bandgap materials, and raised thoughtful concerns regarding the potential impact of the second-order Doppler shift, certain aspects of the quantum chemical calculations, and apparent discrepancies with experimental data. These were valuable points that helped strengthen the manuscript. In my assessment, the authors have addressed them in a convincing and sufficient manner. The minor discrepancies between the experimental results and the quantum chemical calculations do not, in my view, diminish the high impact and broad interest of the central experimental achievement reported in this work.

I therefore recommend the manuscript for publication in Nature.

REFEREES #1 & #2 (REMARKS TO THE AUTHOR)

Recent achievement of a major milestone in the topical research field aimed at studies of the uniquely low energy state (8.4 eV) nuclear isomer, namely, its laser excitation, led to an explosive development of the field. This fast progress indicates that highly anticipated applications of this isomer, such as building the Mössbauer nuclear clock, development of super sensors for material studies and search for dark matter and time-dependence of the fundamental constants, are now right on the horizon. This paper opens a new avenue in experimental investigation of the ^{229}Th solid compounds, significantly broadening the scope of the current research in this field. While, so far, such studies have been focused on the ^{229}Th doped metal fluoride crystals with the band gap exceeding the isomer energy (8.4 eV), this work for the first time experimentally investigates laser excitation of ^{229}Th in a narrow band host (~ 6 eV), namely, in $^{229}\text{ThO}_2$. While an internal conversion channel was eliminated or significantly suppressed in the wide band gap materials, it greatly dominates the radiative channel in the narrow band gap materials, reducing the isomer lifetime by 8 orders of magnitude compared to the radiative decay time. Measuring the number of electrons ejected from the ThO_2 surface because of the internal conversion process as the function of frequency detuning of laser radiation from the resonant nuclear transition allows for monitoring the spectrum of the nuclear absorber dependent on the parameters of the host environment. The proof of principle of such novel laser Conversion Electron Mössbauer Spectroscopy (CEMS) has been demonstrated in this work for the first time. The possibility to increase the stability of ^{229}Th nuclear clocks using such narrow band gap solids (compared to the clocks based on the wide gap ^{229}Th doped metal fluoride crystals possessing large ~ 10 kHz inhomogeneous broadening due to dipole-dipole interaction between ^{229}Th and F nuclear spins) by reducing an interrogation time due to accelerated nuclear decay has been also outlined. All these three mutually related factors (new class of materials, CEMS and a way to achieve high stability) point together to the breakthrough nature of this research, qualifying it for publication in Nature. At the same time, several comments and questions need to be addressed in the paper.

- CEMS has been widely used by Mössbauer community for over 50 years as a sensitive tool for studying the nuclear environment (see for example, M. J. Tricker, J.M. Thomas, A.P. Winterbottom, Conversion-electron Mössbauer spectroscopy for the study of solid surfaces, *Surface Science*, 45, 601, 1974). We believe this fact needs to be acknowledged together with the novelty introduced by this work into this technique, namely, using the laser source instead of the radioactive sources used so far.

We thank the referees for pointing out that we did not adequately explain the long history of CEMS. To that end we have added a sentence referencing its history as well as its recent application in the observation of the ^{45}Sc isomer:

“Conversion electron Mössbauer spectroscopy (CEMS), which has been used for decades as an important materials probe [1–3] and was recently used to detect the 12.4 keV nuclear clock transition in ^{45}Sc [4], has been proposed for detection of ^{229}Th nuclear excitation [5, 6].”

- In connection with the above comment an important question arises. The ability to record the spectral line of the nuclear absorber by means of CEMS relies on using spectrally narrow source (compared to the spectral width of the absorber). However, the linewidth of a laser used in this work was a million times broader than the linewidth of the nuclear transition. It seems not much spectral information can be recovered from this type of measurement except the resonance position. The full potential of this technique can be realized in the future when sufficiently narrow sources (such as based on intracavity HHG) become available.

The referees are correct in pointing out that a great deal more information about the chemical environment will become available as the excitation laser becomes narrower. This indeed will be the direction of future work.

- It is emphasized several times that CEMS, unlike fluorescence spectroscopy, allows to study materials whose work function is less than the nuclear transition energy. It may be true for the specific case of ^{229}Th compounds, where an internal conversion factor is enormously high ($\sim 10^8$ - 10^9) so that the fluorescence is too weak to be registered. But in general, both fluorescence and CEMS are widely used to study materials whose work function is less than the nuclear transition energy. A common example is ^{57}Fe where an internal conversion factor is only 9. The other example is ^{45}Sc , where an internal conversion factor is ~ 400 . Perhaps, this point needs to be clarified to avoid misunderstanding.

We thank the referees for pointing out this subtly. As noted, for ^{229}Th , IC decay can be expected to dominate radiative decay when it is allowed, however, in other nuclei where the nuclear energy is less well matched to electronic energies both processes can occur. To make this clearer, we have changed the second paragraph to emphasize that the internal conversion coefficient is the root cause of the impossibility to perform combined CEMS and fluorescence spectroscopy of the ^{229}Th isomer:

“This leads to the requirement that the host material possess a band gap larger than the isomeric transition energy ($> E_{\text{iso}}$), since the large internal conversion coefficient of the isomeric state ($\sim 10^8$ - 10^9 [7, 8]) extinguishes the nuclear fluorescence.”

- It would be beneficial for the readers to mention the recent work on resonant excitation of the long-lived ^{45}Sc nuclear isomer (Yu. Shvydko, et al., Resonant X-ray excitation of the nuclear clock isomer ^{45}Sc , Nature 622, 471 (2023)). Indeed, in addition to ^{229}Th it is the only one more potential nuclear clock isomer which was directly resonantly excited with x-ray laser. Besides, similar to this work, the technique used for detection of the resonant excitation of ^{45}Sc was based on the internal conversion process, though instead of direct counting the conversion electrons, the characteristic K_α and K_β x-ray photons at ~ 4 keV caused by internal conversion process were recorded.

We agree and have now include a reference to this important work in the first sentence of the fourth paragraph:

“Conversion electron Mössbauer spectroscopy (CEMS), which has been used for decades as an important materials probe [1–3] and was recently used to detect the 12.4 keV nuclear clock transition in ^{45}Sc [4], has been proposed for detection of ^{229}Th nuclear excitation [5, 6].”

- The statement about improvement of nuclear clock stability by reducing interrogation time due to reduction of T1 by an internal conversion process in the narrow band gap materials, pointed out several times in the text, may cause confusion and need clarification. Indeed, based on this statement one may conclude that using such materials is the most advantages for producing stable nuclear clock, which is in an obvious contradiction with a common sense to use quantum transitions from the long-lived states (and hence narrowest possible linewidth) to maximize clock’s accuracy and stability. Apparently, using narrow band gap materials for nuclear clock (resulting in the life-time line broadening to 16 kHz) is more profitable than using a wide band gap materials with large inhomogeneous line broadening (such as metal fluorides with 10 kHz linewidth). But it is not beneficial at all as compared to using the wide band gap spinless materials with very narrow spectral lines (which were discussed in the work of the same group [4]) because Q factor in these materials would be dramatically higher than in the narrow band gap materials resulting in higher clock stability.

We thank the referees for clarifying this subtle point and realize we should have provided a more thorough explanation. We agree with the referees’ interpretation. In the high band gap fluorides, the transition linewidth is predicted to be as small as roughly 1 kHz, due primarily to magnetic dipole broadening due to the copious F atoms; measurements so far have recorded linewidths about an order of magnitude higher and it is currently unclear if those are laser limited or reflect the contribution of other broadening mechanisms like crystal strain. Regardless, when a transition is inhomogeneously broadened the clock interrogation cycle can be decreased to roughly $1/(\text{broadened linewidth})$ without degrading the measured Q , and this increased interrogation rate improves the clock stability. Therefore, compared to the metal fluorides the CEMs-based ThO_2 appears attractive.

When applied to the spinless host material, $\text{Th}(\text{SO}_4)_2$, this analysis changes. There, the linewidth could be so much narrower (i.e. $\sim \text{Hz}$) that a radiative decay readout is preferred. However, it could be that even there the technical challenge of maintaining coherence in the local oscillator for tens to hundreds of seconds will lead to a preference for the a CEMS-based clock. However, given that $\text{Th}(\text{SO}_4)_2$ is unproven experimentally and the systematics in such a material are unexplored, we prefer not to speculate. Instead, to make this clearer to the reader, the introductory discussion of the CEMS clock has been changed to simply state the estimated performance and leave it to the reader to compare to other proposals

“In addition to providing a means to study low band gap materials containing ^{229}Th , CEMS could also allow significant improvements in nuclear clock stability as the IC decay rate is roughly 10^8 times faster than the radiative decay rate, enabling a much faster clock interrogation rate. This rapid interrogation reduces the stability demands on the local oscillator while providing a projected clock instability as low as $\sim 10^{-18}$ at 1 s.”

- We believe the statement “Further, because ThO_2 can be made from spinless isotopes [4] and the internal conversion decay process reduces the isomeric state lifetime to only $\sim 10 \mu\text{s}$, allowing $\sim 10^8$ relative reduction in clock interrogation time, a conversion-electron-based nuclear clock could lead to a $\sim 10^4$ improvement in clock

instability” needs to be either explained or corrected. Indeed, the role of spinless host in this context is not clear. Once the linewidth is determined by the life-time broadening (~ 16 kHz), any narrow band gap host (spinless or not) would lead to the same result (i.e., stability improvement compared to the wide band gap materials with similarly large inhomogeneous broadening due to quenching of nuclear decay).

The referees are correct that changing from a spinful to spinless material has a much less dramatic effect for CEM-based clock than it would in a large band gap crystal using fluorescence readout, since the homogeneous and inhomogeneous broadenings are convolved. Nonetheless, constructing the CEM-based clock out of spinless isotopes does limit additional inhomogeneous broadening. Therefore, we have chosen to point out that ThO₂ is spinless to aid the reader that wants to estimate the clock performance themselves. To make this clearer we have change the word ‘because’ to ‘given’ to clarify that this property is needed to estimate the performance but is not the only driving factor.

“Further, given ThO₂ can be made from spinless isotopes [9] and the internal conversion decay process reduces the isomeric state lifetime to only ~ 10 μ s, allowing $\sim 10^8$ relative reduction in clock interrogation time, a conversion-electron-based nuclear clock could lead to a $\sim 10^4$ improvement in clock instability.”

Referees: Olga Kocharovskaya and Xiwen Zhang

REFEREE #3 (REMARKS TO THE AUTHOR)

The manuscript by Elwell and coworkers presents a significant and timely experimental advance: the use of a VUV laser to perform ²²⁹Th Conversion Electron Mössbauer Spectroscopy (CEMS) on ThO₂. The authors correctly highlight the primary challenge of performing nuclear resonance spectroscopy in materials where the nuclear transition energy (8.3 eV) is greater than the material’s band gap (in this case ~ 6 eV). The successful application of laser-driven CEMS to overcome this fundamental obstacle is a commendable achievement and represents a valuable contribution to the field. The manuscript makes two high-impact claims: (1) that this technique opens the door to studying the ²²⁹Th transition in a new class of low-bandgap materials, and (2) that this approach could lead to a substantial improvement in nuclear clock stability. The first claim is well-supported and compelling. The demonstrated ability to acquire a clean spectrum where traditional methods would fail is a clear strength of this work. However, the second, and arguably more significant, is that the CEMS approach can lead to a higher sampling rate for these types of clocks and thus improvement in stability over other proposed Th-based clocks. However, I think immediate comparisons are premature. Bandgaps tend to correlate with Debye temperatures so materials amenable to CEMS (low bandgap) may then have Debye temperatures low enough that second order doppler shifts become problematic towards clock stability whereas conversely high band-gap materials do not face this issue as significantly. Establishing the magnitude of SODS in Thoria is outside of the scope of this manuscript but the concept should at least be addressed in the context of the atomic clock performance to prevent any misleading of the readers. The recent PRL 134,113801 (2025) from Higgins et al. working on understanding the second order doppler effect towards a Th-229 atomic clock touches directly on this topic. I think this latter publication should be cited in this manuscript and discussed in this context. Arguably it could be made to support the case here. Moreover, I think the calculations in the manuscript need substantially more experimental validation and the evidence offered in the manuscript towards this end is nearly misleading. Overall, I think the experimental work is heroic but specialized. The mathematical derivations are sound, but the quantum chemical calculations are poorly validated. I think there is substantial impact limited to a smaller community than the general Nature readership. Therefore, I believe these authors would better reach their target audience in another journal. Below are specific points that I think if addressed would strengthen the manuscript.

We thank the referee for review of our manuscript. As detailed in the responses below, we have addressed the concerns raised regarding the validity of the work and thereby have made the manuscript much clearer. Therefore, we ask that the referee please reconsider their opinion about the impact of the work and join the other three referees who recommend publication in Nature. We believe this work, which potentially opens a new field of research, is best served with the wide readership of Nature.

- As noted above, the discussion of clock stability is incomplete without a thorough treatment of the Second-Order Doppler Shift. The authors should address the topic of SODS as a potential source of frequency instability. They should provide an estimate for the SODS-induced temperature sensitivity in this system and discuss the demanding technical requirements for its stabilization. This will provide a more realistic assessment of the proposed clock’s potential for the readers.

We thank the referee for bringing attention to the fact that readers will want more detail on the non-leading order broadenings and shifts for the clock system. Additional discussion of the second-order Doppler shift and isomeric shifts have been added to the discussion of clock’s stability (indicated in bold):

“For such a clock, assuming $^{229}\text{ThO}_2$ is produced from ^{16}O , the largest sources of instability would be broadening due to magnetic dipole interactions between the ^{229}Th nuclei and lifetime broadening. Given that the Th-Th distance in ThO_2 is 3.96 \AA [10], the expected Zeeman broadening due to neighboring ^{229}Th nuclei is $\sim 10 \text{ Hz}$. **The next largest sources of broadening would be the varying intrinsic isomeric shift and second-order Doppler shift arising from temperature gradients across the ThO_2 sample. Drawing estimates from other Mössbauer experiments, typical intrinsic isomeric shifts range from 0.1-5 kHz/K [11–14], as supported by recent measurements of the intrinsic isomeric shift in $^{229}\text{Th}:\text{CaF}_2$ at 0.4 kHz/K [15]. Within the Debye model, the second-order Doppler shift can be estimated to be $\lesssim 1 \text{ Hz/K}$ at 4K, using the lowest reported value of 236 K for the Debye temperature of ThO_2 [16]. With temperature gradients stabilized across the sample to $\leq 0.1 \text{ mK}$, the intrinsic isomeric and second-order Doppler shifts result in a broadening of roughly 0.5 Hz. Thus, the primary source of instability is the lifetime broadening of $\Gamma \approx 2\pi \times 16 \text{ kHz}$.”**

- The calculations need more experimental validation. The authors mention the 6.18 eV calculated bandgap is in good agreement with literature values but the bandgap from Mock et al. cited in this manuscript is substantially lower (5.4 eV) than the calculated bandgap here. What in the calculation is the source of this difference? The authors argue their calculated frequency dependent dielectric functions are comparable to Ref. 37 and thus experimentally validate the calculations, but inspection of Fig 4 in that reference demonstrates substantial qualitative differences that require explanation or alternative validation of these calculations. The other experimental validation offered is the absorption coefficient of 0.1 nm^{-1} calculated in agreement with ellipsometric measurements from figure 3.4 in a seemingly non-peer reviewed capstone project write-up. That figure has four measurements in it giving a distribution of values arguably around 0.1 nm^{-1} the spectra also lack any of the qualities in the calculated spectrum of Figure 6 in this manuscript in this same region. I don’t diminish the reference being a capstone project but there must be a better reference to ground these calculations in or an explanation resolving why there are so many differences from the spectra calculated here. Even comparing the geometry optimized structure to the actual crystal structure would be a meaningful step in the right direction.

We agree with the referee that it is important to validate the computational work as much as possible, even though the goal of theory for this work is to provide a qualitative understanding of this new system. Towards that, we have updated our “best” band gap calculation to 6.20 eV so that the band gap comes from the same calculation as the absorption spectrum. We also provide references to a range of experimental results that report band gaps from 5.4 eV to 6.9 eV, with most recent experimental data finding $\sim 5.9 \text{ eV}$. Further, in the SI we have also noted (with relevant citations) that many attempts have been made to measure the band gap of ThO_2 , resulting in a spread of values that reflect the complexity of ThO_2 in terms of defects and morphology. Those may be the origin of discrepancies between measured and computed band gaps, particularly since the measured values are extracted from Tauc plots and are therefore subject to some error. Given the difficulties in measuring and calculating band gaps, we feel this offers sufficient validation of our GW calculations, especially since the goal in this work is a qualitative understanding of this new system.

On the subject of the dielectric function, quantifying the agreement between theory and experiment is difficult as the real and imaginary parts have multiple features over the considered energy range. Important features are seen in both the measured and computed functions: the real part rises from its low-energy value of 4 to a peak at around 6 eV and then falls, with some secondary peaks and shoulders, to a value of about 2 at 8-9 eV; the imaginary part is small at low energies, rising steeply around 5 eV to a plateau. Two qualitative differences can be seen in the imaginary part ϵ_2 : in the measured function the rise at 5 eV is more gentle so the plateau is reached at 8 eV, while in the computed function the rise is sharp so the plateau is reached at 6 eV (note that experiment and theory agree on the value of ϵ_2); and the experimental ϵ_2 has a low-energy “tail” such that it is non-zero down to 1-2 eV, while the computed ϵ_2 is zero below 3-4 eV. These differences are likely to come from defects in the measured crystal - this point is made in the paper by Mock *et al.* to explain the low-energy tail. We have adjusted the text such that the match is described as “good” rather than “excellent” and the differences in the imaginary part are attributed to crystalline defects.

In summary, while there are some imperfections in the match between the computed and experimental dielectric functions, we believe that it is adequate, and indeed common in the field, to give us confidence in our calculation of the absorption coefficient at 8.4 eV. This is the value compared to the measurement in the thesis by Gillis.

Unfortunately, we have not been able to find an equivalent measurement in a peer-reviewed journal article; most absorption studies do not go to such short wavelengths, so far above the band gap energy of ThO_2 .

The match between the computed and experimental lattice parameters (5.617 Å and 5.597 Å respectively) has also been added to the computational methods section as validation of the optimization method.

- Along these lines, the authors use a 500 eV plane wave cutoff for the geometry optimization but then use a 400 eV for the GOW0 calculations. Table IV might indicate a reason for this choice, but it should be more plainly justified – or the two calculations should be done with the same cutoff.

We apologize for the unclear reporting of computational details in the SI. We now report the band gap as 6.20 eV, rather than 6.18 eV, because this value was computed with a 500 eV plane wave cutoff. We agree that the convergence test in Table IV shows that changing the cut-off from 400 eV to 500 eV does not affect the results significantly. The *GW* band gap and *GW*+BSE absorption spectrum are now reported with identical parameters. 400 eV is noted as the cut-off that was used for convergence testing of other settings.

- I find the thought of doing Th-229 nuclear resonance vibrational spectroscopy at sub-meV resolution a very good idea and this group should pursue that in the future.

We appreciate the referee’s encouragement, and this will definitely be an avenue of further researcher when higher spectral density VUV laser sources become available.

- Figure 8 features no x-axis label.

We thank the referee for catching this error. The figure has been replotted.

REFEREE #4 (REMARKS TO THE AUTHOR)

The ^{229}Th nucleus is a prime candidate for developing nuclear optical clocks and probing new physics beyond the Standard Model. Current approaches for building nuclear optical clocks include: (i) The ionic approach – based on trapped ^{229}Th ions, which may achieve the highest precision but suffers from extremely weak nuclear excitation and fluorescence signals (not yet observed experimentally) due to the limited number of trapped ions (~ 1000). (ii) The doped-crystal approach – based on ^{229}Th -doped crystals (e.g., $\text{Th}:\text{CaF}_2$ and $\text{Th}:\text{LiSrAlF}_6$), which provides stronger signals owing to the large number of doped nuclei but the precision may be limited by electronic and phononic interactions in the crystal. (iii) The thin-film approach – based on $^{229}\text{ThF}_4$ thin films, which requires less ^{229}Th material but the clock performance may depend sensitively on film geometry. Both the doped-crystal and thin-film approaches require host materials with high band gaps (> 8.4 eV) and vacuum ultraviolet (VUV) transparency. All existing approaches also face the challenge of long clock interrogation times.

This manuscript experimentally demonstrates a new and promising approach based on laser-induced internal-conversion (IC) electrons in $^{229}\text{ThO}_2$ thin films, the band gap of which is smaller than the nuclear transition energy. In contrast to $^{229}\text{ThF}_4$ thin films, the isomeric state in $^{229}\text{ThO}_2$ decays much faster to the ground state via IC, promoting valence-band electrons to the conduction band and ultimately ejecting electrons that can be detected.

Key experimental findings include: (i) A measured nuclear transition frequency of 2020407 GHz, consistent with previous laser excitation experiments in $^{229}\text{Th}:\text{LiSrAlF}_6$. (ii) A reduced isomeric-state lifetime of ~ 10 μs , much shorter than the radiative decay lifetime, enabling a significant reduction in clock interrogation time. The authors also present a comprehensive theoretical analysis of the IC decay rate, including electronic-structure calculations, hyperfine interactions, and isomer shifts in solids.

The results are exciting and represent a significant step forward in nuclear clock development. The concept of an IC-based solid-state nuclear clock is demonstrated to be both architecturally sound and technically feasible. This approach has great potential for the miniaturization of nuclear clocks and for enabling practical applications. Given its expected high impact and broad interdisciplinary interest, I recommend publication in *Nature* after the following questions and comments are addressed.

- In the abstract: “this technique is compatible with materials whose work function is less than the nuclear transition energy...” – please provide the definition and value of the work function of the ThO_2 thin film. In Fig. 3(c), the authors present the calculated Th PDOS, and the energy gap between the top of the conduction band and the Fermi level is clearly larger than 8.4 eV.

We thank the referee for drawing attention to this sentence. This was a sloppily word choice. The more appropriate term is the *ionization energy* of ThO_2 , or the energy difference between the top of the valence

band and vacuum. Due to sensitivity to surface conditions, and the necessity to interpret measurements from thermionic emission and XPS, the ionization energy can only be estimated to be $\sim 6.7 - 9$ eV (such an analysis can be found in [17]). However, for our experiment its exact value is not relevant, as it is absorbed into the electron extraction efficiency as described in the Supplemental Information. Further, conduction band states a few eV from the bottom of the band are known to not stay bound in ThO₂, allowing for tunneling into the vacuum.

In the abstract, in the sentence that originally mentioned the work function, we have made the following change:

Unlike fluorescence spectroscopy of the ²²⁹Th isomeric transition, this technique is compatible with materials whose band gap is less than the nuclear transition energy, opening a wider class of systems to study.

In the third paragraph of the introduction the mention of the work function barrier has been replaced by the following:

If this promoted IC electron originates in a shallow-enough state in the valence band, the electron can overcome the ionization energy barrier and emerge from the material surface (by ionization energy we mean the difference between the top of the valence band and the vacuum [18]).

And in the discussion of the electron extraction efficiency of the SI section “EXPECTED INTERNAL CONVERSION SIGNAL” the mention of the work function has been replaced by:

The extraction efficiency represents the probability that an IC electron is able to leave the ²²⁹ThO₂ target, and combines many physical processes such as whether the electron is promoted high enough into the conduction band to overcome the **ionization energy barrier (either by being in a state above the the ionization energy or tunneling)**, whether the electron inelastically scatters, whether surface conditions are favorable, etc.

- It is unclear whether the observed electron emission follows: (a) IC directly promoting valence-band electrons to the conduction band, followed by escape to vacuum, or (b) photoexcitation of electrons to the conduction band by the laser pulse, followed by IC promotion of these conduction-band electrons into vacuum. The manuscript discusses only process (a). Are there experimental data or theoretical arguments ruling out process (b)?

This is a very interesting question. The process (b) the referee is describing would require the photo-excited electrons to remain excited for several microseconds, however recombination lifetimes for carriers excited to the conduction band are typically sub-nanosecond. See a discussion of the NEET processes in solid-state hosts in PRL 134, 253801 (2025). As such only process (a) was considered. It is intriguing to consider the possibility of intentionally exciting conduction or defect electrons after nuclear excitation to purposefully quench the nuclear transition. To make this point, we have added a sentence to the paper that reads:

It may also be possible to further enhance this performance by purposefully exciting defect or conduction band electrons to realize an ‘on-demand’ quench of the nuclear excitation [19, 20].

- The derivation of the IC rate is comprehensive; however, some of the approximations used require further explanation. In the second paragraph of Page 14, the authors state that “we shall neglect the contribution cross terms in $|W_{c\sigma_c v\sigma_v}^{ge}(k)|^2$ ”, which ultimately alters the order of summation and modulus square operations. This approximation is not obviously reasonable. Correspondingly, the calculated IC rate is about 1.3×10^4 s⁻¹, which is lower than the observed IC decay rate of approximately 10^5 s⁻¹. Does the neglect of the cross terms lead to a significant reduction in the calculated IC rate? This assumption needs to be evaluated in the calculation.

This is an important question. To properly account for the cross-term contributions, one would need to explicitly compute the expansion coefficients themselves. This involves computing all-electron wavefunctions. We are actively pursuing this research direction. For this paper we chose to simply estimate the IC rate without including the cross terms, since their unknown phase difference would lead to some cancellation. This was done because we were being conservative in our attempt to match theory and experiment. However, the referee makes a strong point, namely, the inclusion of these cross terms, without cancellation, allows us to estimate a range of IC rates. Therefore, we now also perform a calculation of the IC rate using the cross terms without cancellation, which results in an estimated lifetime of 30 μ s. We have updated the main text accordingly at the end of the section “Calculation of Isomer Shifts and Internal Conversion Lifetime in ThO₂”.

Sources of the disparity between the measured and theoretical lifetimes could be due to neglecting cross- and higher-order expansion terms neglected in our evaluation of the rate (1) and errors in the local projections of

the plane wave orbitals done by VASP [21]. For example, adding the cross terms coherently may increase the IC rate and thus reduce the estimated IC lifetime to $\sim 30 \mu\text{s}$. Relativistic contraction $\sim (Z\alpha)^2$ may increase the non-relativistic PDOS computed with VASP as much as 40%, so the increase in the IC rate may be up to a factor of $1.4^2 \approx 2$, further reducing the estimated IC lifetime to $\sim 15 \mu\text{s}$. See SI for more details. Small changes to the valence and conduction states expected with the addition of spin-orbit coupling in solid-state calculations and deviations of the sample from bulk $^{229}\text{ThO}_2$ due to both surface and self-radiation damage effects may further affect the IC rate. Compared to the pristine $^{229}\text{ThO}_2$, the IC rate for ^{229}Th adjacent to point defects can be enhanced by the electric quadrupole HFI contribution [22] due to symmetry breaking. Evaluating these uncertainties requires a much more comprehensive study that will be carried out in future work. Here, we simply note that the order-of-magnitude experiment-theory agreement supports the physical interpretation that observed IC decay results from the nucleus relaxing via transferring an electron from an oxide anion into the Th $6p$ component of the conduction band.

Sources of the disparity between the measured and theoretical lifetimes could be due to neglecting cross- and higher-order expansion terms neglected in our evaluation of the rate (1) and errors in the local projections of the plane wave orbitals done by VASP [21]. For example, adding the cross terms coherently reduces the estimated IC lifetime to $\sim 30 \mu\text{s}$. Relativistic contraction $\sim (Z\alpha)^2$ may increase the non-relativistic PDOS computed with VASP as much as 40%, leading to a further reduction of the IC lifetime by a factor of ~ 2 . Together, these would reduce the IC lifetime to $\sim 15 \mu\text{s}$ (see SI for more details). Small changes to the valence and conduction states expected with the addition of spin-orbit coupling in solid-state calculations and deviations of the sample from bulk $^{229}\text{ThO}_2$ due to both surface and self-radiation damage effects may further affect the IC rate. Compared to the pristine $^{229}\text{ThO}_2$, the IC rate for ^{229}Th adjacent to point defects can be enhanced by the electric quadrupole HFI contribution [22] due to symmetry breaking. Evaluating these uncertainties requires a much more comprehensive study that will be carried out in future work. Nonetheless, the order-of-magnitude agreement with experiment supports the physical interpretation that the observed IC decay results from the nucleus relaxing via transferring an electron from an oxide anion into the Th $6p$ component of the conduction band.

This is also further explained at the end of the the updated “INTERNAL CONVERSION RATE DERIVATION” section in the SI:

In obtaining the estimate (23), we have neglected the contribution from the cross terms in the expansion of $|W_{\sigma_c v \sigma_v}^{ge}(\mathbf{k})|^2$. Neglecting off-diagonal HFI matrix elements, the contribution from these terms reads

$$\begin{aligned} \Gamma_{\text{IC}}^{\text{cross-terms}} &\approx \frac{\pi}{3\hbar} \frac{1}{2I_e + 1} \langle g || \mathcal{M} || e \rangle^2 \sum_{njl \neq n'j'l'} \langle njl || \mathcal{T} || njl \rangle \langle n'j'l' || \mathcal{T} || n'j'l' \rangle \\ &\times \sum_{\sigma_c \sigma_v} \left[\frac{2\Omega}{(2\pi)^3} \int_{\text{BZ}} d^3k a_{njl}^*(\mathbf{c}\mathbf{k}\sigma_c) a_{njl}(\mathbf{v}\mathbf{k}\sigma_v) a_{n'j'l'}(\mathbf{c}\mathbf{k}\sigma_c) a_{n'j'l'}^*(\mathbf{v}\mathbf{k}\sigma_v) \delta(\varepsilon_{c\mathbf{k}} - \varepsilon_{v\mathbf{k}} - \hbar\omega_{\text{nuc}}) \right]. \end{aligned} \quad (24)$$

Heuristically, one expects the different terms in Eq. (24) to have different phases and thus their total sum to be small. A more detailed calculation where the expansion coefficients $a_{njl}(\mathbf{c}\mathbf{k}\sigma_c)$ and $a_{njl}(\mathbf{v}\mathbf{k}\sigma_v)$ are computed explicitly is needed to precisely gauge the accuracy of this approximation. This is the subject of a future work. Here, we assume that the expansion coefficients have the same phase (and may thus be taken as all real) and estimate the maximum contributions from the cross terms. In this case, the quantity in the square bracket of Eq. (24) is a “cross-PJDOS” which is obtained by summing over the delta functions weighted by the product of square roots of $|a_{njl}|^2$. The maximum values for the cross terms are listed in Table III. The sum of these cross contributions amounts to

$$\Gamma_{\text{IC}}^{\text{max cross-terms}} \approx 2.6 \times 10^4 \text{ s}^{-1}. \quad (25)$$

The estimated maximum IC rate is therefore

$$\Gamma_{\text{IC}}^{\text{max}} \approx 3.8 \times 10^4 \text{ s}^{-1}, \quad (26)$$

corresponding to an IC lifetime of $30 \mu\text{s}$. A further reduction in the IC lifetime is very plausible if one notes that the relative error in making the non-relativistic approximation $|a_{n(l-1/2)l}(\mathbf{c}\mathbf{k}\sigma_c)|^2 = |a_{n(l+1/2)l}(\mathbf{c}\mathbf{k}\sigma_c)|^2 = |a_{nl}|^2/2$ is $\sim (\alpha Z)^2$, i.e., about 40% for $Z = 90$ of Th. Since relativistic contraction generally increases the electron density near the origin, the IC rate including relativistic effects may be larger than its non-relativistic counterpart by a factor of $\sim 1.4^2 = 2$, thus further reducing the estimated IC lifetime to $\sim 15 \mu\text{s}$.

- Furthermore, in the first paragraph of Page 15, the expansion coefficients are set as $|a_{nl-1/2l}|^2 = |a_{nl+1/2l}|^2 = |a_{nl}|^2/2$. This approximation is introduced to simplify the treatment of spin-orbit coupling. How does it quantitatively affect the calculated IC decay rate? Please provide a justification of its validity.

As with the cross terms, answering this question conclusively requires explicit computation of the a coefficients, which is the focus of ongoing work. Generally speaking, however, the relative error in making this non-relativistic approximation is $\sim (\alpha Z)^2$, i.e, about 40% for $Z = 90$ of Th. Translated to the IC rate, this may mean a factor of $1.4^2 \approx 2$ difference. To address this important point, we have amended the text as described in the updated “Calculation of Isomer Shifts and Internal Conversion Lifetime in ThO₂” section and in the “INTERNAL CONVERSION RATE DERIVATION” section in the SI that are discussed in response to the previous point.

- The authors predict a projected clock instability of about 2×10^{-18} at 1 s averaging. Please provide a more detailed evaluation in the Supplemental Information.

We agree this is an important missing detail. A new SI section has been added under “Determination of Clock Stability”:

“We estimate the clock stability by [23]

$$\sigma = \frac{1}{2\pi QS} \sqrt{\frac{T_e + T_c}{\tau}}, \quad (1)$$

where $Q = f_0/\Delta f$ is the transition quality factor, S is the signal-to-noise ratio (SNR), T_e is the excitation time, T_c is the electron collection time, and τ is the averaging time. As stated in the text, we assume that Δf is dominated by the homogeneous lifetime broadening due to internal conversion and use $\Delta f = 16$ kHz. For our analysis, we assume $T_e = T_c = 12 \mu\text{s}$, and a shot noise limited SNR given by $S = \sqrt{N_{\text{det}}}$, where N_{det} is given by

$$N_{\text{det}} = N_e \times (1 - e^{-T_c/\tau_{\text{IC}}}), \quad (2)$$

and N_e ($\sim 2.5 \times 10^7$) is the total number of ²²⁹Th nuclei excited, $\eta = 0.5$ is the electron detection efficiency in the clock system, and $\tau_{\text{IC}} = 12 \mu\text{s}$. N_e is calculated assuming that the probe laser linewidth is significantly smaller than the lifetime limited transition linewidth, and the probe laser power is 100 μW . From this we obtain a projected clock performance of $\sim 2 \times 10^{-18}/\sqrt{\tau}$.”

-
- [1] M. J. Tricker, J. M. Thomas, and A. Winterbottom, Conversion-electron mössbauer spectroscopy for the study of solid surfaces, *Surface Science* **45**, 601 (1974).
 - [2] O. Massenet, Conversion electron mössbauer spectroscopy applied to magnetic film and surface studies, *IEEE Transactions on Magnetism* **18**, 705 (1982).
 - [3] T. Mitsui, K. Fujiwara, S. Sakai, S. Li, J. Okabayashi, Y. Kobayashi, and M. Seto, Synchrotron based conversion electron mössbauer spectroscopy, *Interactions* **245**, 347 (2024).
 - [4] Y. Shvyd'ko, R. Röhlberger, O. Kocharovskaya, J. Evers, G. A. Geloni, P. Liu, D. Shu, A. Miceli, B. Stone, W. Hippler, B. Marx-Glowna, I. Uschmann, R. Loetzsch, O. Leupold, H. C. Wille, I. Sergeev, M. Gerharz, X. Zhang, C. Grech, M. Guetg, V. Kocharyan, N. Kujala, S. Liu, W. Qin, A. Zozulya, J. Hallmann, U. Boesenberg, W. Jo, J. Möller, A. Rodriguez-Fernandez, M. Youssef, A. Madsen, and T. Kolodziej, Resonant X-ray excitation of the nuclear clock isomer ⁴⁵Sc, *Nature* **622**, 471 (2023).
 - [5] L. von der Wense et al., A laser excitation scheme for ²²⁹mTh, *Phys. Rev. Lett.* **119**, 132503 (2017).
 - [6] L. C. von der Wense, B. Seiferle, C. Schneider, J. Jeet, I. Amersdorffer, N. Arlt, F. Zacherl, R. Haas, D. Renisch, P. Mosel, P. Mosel, M. Kovacev, U. Morgner, C. E. Düllmann, E. R. Hudson, and P. G. Thirolf, The concept of laser-based conversion electron Mössbauer spectroscopy for a precise energy determination of ²²⁹mTh, *Hyperfine Interactions* **240**, 23 (2019).
 - [7] V. Strizhov and E. Tkalya, Decay channel of low-lying isomer state of the ²²⁹Th nucleus. possibilities of experimental investigation, *Sov. Phys. JETP* **72**, 387 (1991).
 - [8] E. V. Tkalya, C. Schneider, J. Jeet, and E. R. Hudson, Radiative lifetime and energy of the low-energy isomeric level in ²²⁹Th, *Phys. Rev. C* **92**, 054324 (2015).
 - [9] H. W. T. Morgan, J. E. S. Terhune, R. Elwell, H. B. T. Tan, U. C. Perera, A. Derevianko, E. R. Hudson, and A. N. Alexandrova, A spinless crystal for a high-performance solid-state ²²⁹Th nuclear clock (2025), [arXiv:2408.12309](https://arxiv.org/abs/2408.12309).
 - [10] J. Belle and R. M. Berman, *Thorium dioxide: properties and nuclear applications*, Tech. Rep. (USDOE Assistant Secretary for Nuclear Energy, Washington, DC. Office of Naval Reactors, 1984).
 - [11] G. M. Rothberg, S. Guimard, and N. Benczer-Koller, Temperature dependence of the β -tin isomer shift, *Phys. Rev. B* **1**, 136 (1970).

- [12] S. T. Lin, G. M. Rothberg, and E. F. Skelton, Temperature dependence of isomer shift of ^{119}Sn in Mg_2Sn and $\beta - \text{Sn}$, *Phys. Rev. B* **10**, 3789 (1974).
- [13] Y. Hazony, $3d$ density distribution and the intrinsic temperature dependence of the mössbauer isomer shift in iron compounds, *Phys. Rev. B* **7**, 3309 (1973).
- [14] H. K. Perkins and Y. Hazony, Temperature-dependent crystal field and charge density: Mössbauer studies of FeF_2 , KFeF_3 , FeCl_2 , and FeF_3 , *Phys. Rev. B* **5**, 7 (1972).
- [15] J. S. Higgins, T. Ooi, J. F. Doyle, C. Zhang, J. Ye, K. Beeks, T. Sikorsky, and T. Schumm, Temperature sensitivity of a thorium-229 solid-state nuclear clock, *Phys. Rev. Lett.* **134**, 113801 (2025).
- [16] C. L. Dugan, C. Young, R. Carmona, M. Schneider, J. C. Petrosky, J. Mann, E. Hunt, and J. W. McClory, The debye temperature of a single crystal thorium–uranium dioxide alloy, *physica status solidi (RRL) – Rapid Research Letters* **12**, 1800436 (2018), <https://onlinelibrary.wiley.com/doi/pdf/10.1002/pssr.201800436>.
- [17] L. von der Wense and C. Zhang, Concepts for direct frequency-comb spectroscopy of ^{229}Th and an internal-conversion-based solid-state nuclear clock, *The European Physical Journal D* **74**, 146 (2020).
- [18] L. Lin, R. Jacobs, T. Ma, D. Chen, J. Booske, and D. Morgan, Work function: Fundamentals, measurement, calculation, engineering, and applications, *Phys. Rev. Appl.* **19**, 037001 (2023).
- [19] J. E. S. Terhune, R. Elwell, H. B. T. Tan, U. C. Perera, H. W. T. Morgan, A. N. Alexandrova, A. Derevianko, and E. R. Hudson, Photoinduced quenching of the ^{229}Th isomer in a solid-state host, *Phys. Rev. Res.* **7**, L022062 (2025).
- [20] F. Schaden, T. Riebner, I. Morawetz, L. T. De Col, G. A. Kazakov, K. Beeks, T. Sikorsky, T. Schumm, K. Zhang, V. Lal, G. Zitzer, J. Tiedau, M. V. Okhapkin, and E. Peik, Laser-induced quenching of the th-229 nuclear clock isomer in calcium fluoride, *Phys. Rev. Res.* **7**, L022036 (2025).
- [21] S. Maintz, V. L. Deringer, A. L. Tchougréeff, and R. Dronskowski, Analytic projection from plane-wave and paw wavefunctions and application to chemical-bonding analysis in solids, *Journal of Computational Chemistry* **34**, 2557–2567 (2013).
- [22] P. V. Bilous, N. Minkov, and A. Pálffy, Electric quadrupole channel of the 7.8 eV Th 229 transition, *Phys. Rev. C* **97**, 2 (2018).
- [23] M. M. Boyd, *High Precision Spectroscopy of Strontium in an Optical Lattice: Towards a New Standard for Frequency and Time*, *Ph.D. thesis*, University of Colorado, Boulder (2007).